# PETIMOT: A Novel Framework for Inferring Protein Motions from Sparse Data Using SE(3)-Equivariant Graph Neural Networks

## Abstract

Proteins move and deform to ensure their biological functions. Despite significant progress in protein structure prediction, approximating conformational ensembles at physiological conditions remains a fundamental open problem. This paper presents a novel perspective on the problem by directly targeting continuous compact representations of protein motions inferred from sparse experimental observations. We develop a task-specific loss function enforcing data symmetries, including scaling and permutation operations. Our method PETIMOT (Protein sEquence and sTructure-based Inference of MOTions) leverages transfer learning from pre-trained protein language models through an SE(3)-equivariant graph neural network. When trained and evaluated on the Protein Data Bank, PETIMOT shows superior performance in time and accuracy, capturing protein dynamics, particularly large/slow conformational changes, compared to state-of-the-art diffusion and flow-matching approaches, as well as traditional physics-based models.

## 1 Introduction

Proteins orchestrate biological processes in living organisms by interacting with their environment and adapting their three-dimensional (3D) structures to engage with cellular partners, including other proteins, nucleic acids, small-molecule ligands, and co-factors. In recent years, spectacular advances in high-throughput deep learning (DL) technologies have provided access to reliable predictions of protein 3D structures at the scale of entire proteomes (Varadi et al., 2024). These breakthroughs have also highlighted the complexities of protein conformational heterogeneity. State-of-the-art predictors struggle to model alternative conformations, fold switches, large-amplitude conformational changes, and solution ensembles (Chakravarty et al., 2025).

The success of AlphaFold2 (Jumper et al., 2021) has stimulated machine-learning approaches focused on inference-time interventions in the model to generate structural diversity. They include enabling or increasing dropout (Raouraoua et al., 2024; Wallner, 2023), or manipulating the evolutionary information given as input to the model (Kalakoti & Wallner, 2024; Wayment-Steele et al., 2023; Del Alamo et al., 2022; Stein & Mchaourab, 2022). Despite promising results on specific families, several studies have emphasised the difficulties in rationalising the effectiveness of these modifications and interpreting them (Porter et al., 2024; Bryant & Noé, 2024). Moreover, these cannot be transferred to protein language model (pLM)-based predictors that do not rely on multiple sequence alignments. Researchers have also actively engaged in the development of deep learning frameworks based on diffusion, or the more general flow matching, to generate conformational ensembles (Lewis et al., 2025; Wang et al., 2025). While being several orders of magnitude cheaper than Molecular Dynamics (MD) simulations, these models remain computationally intensive, require massive MD training data, and are limited to sampling approximate equilibrium distributions.

This work presents a new glance at the protein conformational diversity problem. Instead of learning and sampling from multi-dimensional empirical distributions, we propose to learn eigenspaces (the structure) of the positional covariance matrices in collections of experimental 3D structures and generalize these over different homology levels. The use of experimental structure collections to infer protein dynamics through Principal Component Analysis (PCA) is well-established in the literature (Best et al., 2006; Schneider et al., 2025; Lombard et al., 2024a; Yang et al., 2009). The diversity

present within – even a modest number of – experimental 3D structures of the same protein or close homologs is a good proxy for the conformational heterogeneity of proteins in solution (Best et al., 2006) and can generally be (almost fully) explained by a small set of linear vectors, also referred to as modes (Lombard et al., 2024a; Yang et al., 2009). Moreover, interpolation trajectories performed in PCA space inferred from experimental structures can recapitulate intermediate functional states (Lombard et al., 2024a). Although linear spaces may not be well-suited for capturing highly complex non-linear motions, such as loop deformations, they offer multiple advantages. These include faster learning due to the reduced complexity of the model, improved explainability as the components directly correspond to interpretable data dimensions, faster inference, and the straightforward combination or integration of multiple data dimensions.

To summarize, our main contributions are:

- We provide a novel formulation of the protein conformational diversity problem.
- We present a novel benchmark representative of the Protein Data Bank structural diversity and compiled with a robust pipeline (Lombard et al., 2024a), along with data- and task-specific metrics.
- We develop a SE(3)-equivariant Graph Neural Network architecture equipped with a novel symmetry-aware loss function for comparing linear subspaces, with invariance to permutation and scaling. Our model, PETIMOT, leverages embeddings from pre-trained pLMs, building on prior proof-of-concept work demonstrating that they encode information about functional protein motions (Lombard et al., 2024b).
- PETIMOT is trained on sparse experimental data without any use of simulation data, in contrast with Timewarp for instance (Klein et al., 2024). Moreover, our model does not require physics-based guidance or feedback, unlike (Wang et al., 2025) for instance.
- Our results demonstrate the capability of PETIMOT to generalise across protein families (contrary to variational autoencoder-based approaches) and to compare favorably in running time and accuracy to the physics-based Normal Mode Analysis.

## 2 RELATED WORKS

**Protein structure prediction and generating conformational ensembles.** AlphaFold2 was the first end-to-end deep neural network to achieve near-experimental accuracy in predicting protein 3D structures, even for challenging cases with low sequence similarity to proteins with resolved structures (Jumper et al., 2021). Later works have shown that substituting the input alignment by embeddings from a pLM can yield comparable performance (Lin et al., 2023; Hayes et al., 2024; Weissenow et al., 2022; Wu et al., 2022).

Beyond the single-structure frontier, several studies have underscored the limitations and potential of protein structure predictors (PSP) for generating alternative conformations (Saldaño et al., 2022; Lane, 2023; Bryant & Noé, 2024; Chakravarty et al., 2025). Approaches focused on re-purposing AlphaFold2 include dropout-based massive sampling (Raouraoua et al., 2024; Wallner, 2023), guiding the predictions with state-annotated templates (Faezov & Dunbrack Jr, 2023; Heo & Feig, 2022), and inputting shallow, masked, corrupted, subsampled or clustered alignments (Kalakoti & Wallner, 2024; Wayment-Steele et al., 2023; Del Alamo et al., 2022; Stein & Mchaourab, 2022). Despite promising results, these approaches remain computationally expensive and their generalisability, interpretability, and controllability remain unclear (Bryant & Noé, 2024; Chakravarty et al., 2025). More recent works have aimed at overcoming these limitations by directly optimising PSP learnt embeddings under low-dimensional ensemble constraints (Yu et al., 2025).

Another line of research has consisted in fine-tuning or re-training AlphaFold2 and other single-state PSP under diffusion or flow matching frameworks (Jing et al., 2024; Abramson et al., 2024; Krishna et al., 2024). More generally, diffusion- and flow matching-based models allow for efficiently generating diverse conformations conditioned on the presence of ligands or cellular partners (Jing et al., 2023; Ingraham et al., 2023; Wang et al., 2025; Liu et al., 2024; Zheng et al., 2024). Despite their strengths, these techniques are prone to hallucination.

Parallel related works have sought to directly learn generative models of equilibrium Boltzmann distributions using normalising flows (Lewis et al., 2025; Noé et al., 2019; Klein et al., 2024), or

machine-learning force fields based on equivariant graph neural network (GNN) representations (Wang et al., 2024a), to enhance or replace molecular dynamics (MD) simulations. The BioEmu model (Lewis et al., 2025) predicts large and biologically meaningful conformational changes observed in the Protein Data Bank (PDB) (Berman et al., 2000) and approximates long MD simulations.

**Protein conformational heterogeneity manifold learning.** Unsupervised, physics-based Normal Mode Analysis (NMA) has long been effective for inferring functional modes of deformation by leveraging the topology of a single protein 3D structure (Grudinin et al., 2020; Hoffmann & Grudinin, 2017; Hayward & Go, 1995). While appealing for its computational efficiency, the accuracy of NMA strongly depends on the initial topology (Laine & Grudinin, 2021), limiting its ability to model extensive secondary structure rearrangements. Recent efforts have sought to address these limitations by directly learning continuous, compact representations of protein motions from sparse experimental 3D structures. These approaches employ dimensionality reduction techniques, from classical manifold learning methods (Lombard et al., 2024a) to neural network architectures like variational auto-encoders (Ramaswamy et al., 2021). By projecting motions onto a learned low-dimensional manifold, these methods enable reconstruction of accurate, physico-chemically realistic conformations, both within the interpolation regime and near the convex hull of the training data (Lombard et al., 2024a). Additionally, they assist in identifying collective variables from molecular dynamics (MD) simulations, supporting importance-sampling strategies (Chen et al., 2023a; Belkacemi et al., 2021; Bonati et al., 2021; Wang et al., 2020; Ribeiro et al., 2018). Despite these advances, such approaches are currently constrained to family-specific models.

**E(3)-equivariant graph neural networks.** Graph Neural Networks (GNN) have been extensively used to represent protein 3D structures. They are robust to transformations of the Euclidean group, namely rotations, reflections, and translations, as well as to permutations. In their simplest formulation, each node represents an atom and any pair of atoms are connected by an edge if their distance is smaller than a cutoff or among the smallest $k$ interatomic distances. Many works have proposed to enrich this graph representation with SE(3)-equivariant features informing the model about interatomic directions and orientations (Ingraham et al., 2019; Jing et al., 2020; Dauparas et al., 2022; Krapp et al., 2023; Wang et al., 2024b). To go beyond local 3D neighbourhoods while maintaining sub-quadratic complexity, Chroma adds in randomly sampled long-range connections (Ingraham et al., 2023).

## 3 DATA REPRESENTATION AND PROBLEM FORMULATION

To generate training data, we exploit experimental protein single chain structures available in the PDB. We first clustered these chains based on their sequence similarity. Then, within each cluster, we aligned the protein sequences and used the resulting mapping for superimposing the 3D coordinates (Lombard et al., 2024a). It may happen that some residues in the multiple sequence alignment do not have resolved 3D coordinates in all conformations. To account for this uncertainty, we assigned a confidence score $w_i$ to each residue $i$ computed as the proportion of conformations including this residue. The 3D superimposition puts the conformations' centers of mass to zero and then aims at determining the optimal least-squares rotation minimizing the Root Mean Square Deviation (RMSD) between any conformation and a reference conformation, while accounting for the confidence scores (Kabsch, 1976; Kearsley, 1989),

$$E = \frac{1}{\sum_i w_i} \sum_i w_i (\vec{r}_{ij} - \vec{r}_{i0})^2, \tag{1}$$

where $\vec{r}_{ij} \in \mathbb{R}^3$ is the $i$th centred coordinate of the $j$th conformation and $\vec{r}_{i0} \in \mathbb{R}^3$ is the $i$th centred coordinate of the reference conformation. Next, we defined our ground-truth targets as eigenspaces of the coverage-weighted C$\alpha$-atom positional covariance matrix,

$$C = \frac{1}{m-1} W^{\frac{1}{2}} R^c (R^c)^T W^{\frac{1}{2}} = \frac{1}{m-1} W^{\frac{1}{2}} (R - R^0)(R - R^0)^T W^{\frac{1}{2}}, \tag{2}$$

where $R$ is the $3N \times m$ positional matrix with $N$ the number of residues and $m$ is the number of conformations, $R^0$ contains the coordinates of the reference conformation, and $W$ is the $3N \times 3N$ diagonal coverage matrix. The covariance matrix is a $3N \times 3N$ square matrix, symmetric and real. We decompose $C$ as $C = YDY^T$, where $Y$ is a $3N \times 3N$ matrix with each column defining a

coverage-weighted eigenvector or a principal component that we interpret as a *linear motion*. $D$ is a diagonal matrix containing the eigenvalues. The latter highly depend on the sampling biases in the PDB and thus we do not aim at predicting them.

**Problem formulation.** For a protein of length $N$, let $Y$ be $3N \times K$ *orthogonal* ground-truth deformations,

$$Y^T Y = I_K. \tag{3}$$

Our goal is to find coverage-weighted vectors $X \in \mathbb{R}^{3N \times L}$ whose components $l$ *approximate* some components $k$ of the ground truth $Y$:

$$W^{\frac{1}{2}} \tilde{\mathbf{x}}_l \approx \mathbf{y}_k. \tag{4}$$

Below, we provide three alternative formulations for this problem. PETIMOT's loss function serves two key purposes: it enables effective training of the network to predict subspaces representing multiple distinct modes of deformations – *i.e.*, with low overlap between the subspace's individual linear vectors, while preventing convergence to a single dominant mode.

**The least-square formulation.** To evaluate a predicted motion direction against a ground-truth direction, we use a Least-Square (LS) error, which, together with Mean Absolute Error (MAE), is among the most accepted metrics for regression tasks. Here, we have specifically adapted it to the challenge of evaluating directional motion vectors rather than static coordinates, and scaled between 0 and 1 for better training, interpretability and usability.

For each protein of length $N$ with a coverage $W$, we compute the weighted pairwise *least-square difference* $\mathcal{L}_{kl}$ between ground-truth directions $Y$ and predicted motion directions $X$ for each pair of a $k$ direction in the ground truth and an $l$ direction in the prediction as,

$$\mathcal{L}_{kl} = \frac{1}{N} \sum_{i=1}^{N} \|\vec{y}_{ik} - w_i^{1/2} c_{kl} \vec{x}_{il}\|^2 = \frac{1}{N} \mathbf{y}_k^T \mathbf{y}_k - \frac{1}{N} \frac{(\mathbf{y}_k^T W^{\frac{1}{2}} \mathbf{x}_l)^2}{\mathbf{x}_l^T W \mathbf{x}_l}, \tag{5}$$

where we scaled the ground-truth tensors such that $Y^T Y = N I_K$ and we used the fact that the optimal scaling coefficients $c_{kl}$ between the $k$-th ground truth vector and the $l$-th prediction are given by

$$c_{kl} = \frac{\sum_{i=1}^{N} w_i^{\frac{1}{2}} \mathbf{y}_{ik}^T \mathbf{x}_{il}}{\sum_{i=1}^{N} w_i \mathbf{x}_{il}^T \mathbf{x}_{il}} = \frac{\mathbf{y}_k^T W^{\frac{1}{2}} \mathbf{x}_l}{\mathbf{x}_l^T W \mathbf{x}_l}. \tag{6}$$

This invariance to global scaling is motivated by the fact that we aim at capturing the relative magnitudes and directions of the motion patterns rather than their sign or absolute amplitudes.

**Linear assignment problem.** We then formulate an *optimal linear assignment problem* to find the minimum-cost matching between the ground-truth and the predicted directions. Specifically, we aim to solve the following assignment problem for the least-square (LS) costs,

$$\text{LS Loss} = \min_{\pi \in S_J} \sum_{k=1}^{\min(K,L)} \mathcal{L}_{k,\pi(k)} \tag{7}$$

$$\text{subject to:} \quad \pi : \{1, \dots, \min(K,L)\} \to \{1, \dots, L\}, \quad \pi(k) \neq \pi(k') \text{ for } k \neq k',$$

where $K$ and $L$ are the number of ground-truth and predicted directions respectively, and $\pi(k)$ represents the index of the predicted direction assigned to the $k$-th ground truth direction. This formulation ensures an optimal one-to-one matching, while accommodating cases where the number of predicted and ground-truth directions differs. We backpropagate the loss only through the optimally matched pairs, using *scipy* linear_sum_assignment. We have also tested a smooth version of the loss above with continuous gradients, but it did not improve the performance.

**The subspace coverage formulation.** We propose another formulation of the problem in terms of the subspace coverage metrics (Amadei et al., 1999; Leo-Macias et al., 2005; David & Jacobs,

2011). Specifically, we sum up *squared sinus* (SS) dissimilarities between ground-truth and predicted directions (formally computed as one minus squared cosine similarity),

$$\text{SS Loss} = 1 - \frac{1}{K} \sum_{k=1}^{K} \sum_{l=1}^{K} (\mathbf{y}_k^T W^{\frac{1}{2}} \mathbf{x}_l^{\perp})^2, \tag{8}$$

where the subspace $\{\mathbf{x}_l^{\perp}\}$ is obtained by orthogonalising the coverage-weighted predicted linear subspace $\{W^{\frac{1}{2}} \mathbf{x}_l\}$, where $\mathbf{x}_l^T W \mathbf{x}_l = 1$, using the Gram–Schmidt process. This operation ensures that the loss ranges from zero for identical subspaces to one for mutually orthogonal subspaces and avoids artificially inflating the SS loss due to redundancy in the predicted motions. The order in which the predicted vectors are orthogonalised does not influence the loss, guaranteeing stable training. Appendix A proves this statement. The SS loss is conceptually similar to the comparison of angles between subspaces – see a few recent examples of such subspace comparison from other ML domains in (Zhu et al., 2021; Feng et al., 2023; Chen et al., 2023b; Hawke et al., 2024; Schlaginhaufen & Kamgarpour, 2024).

**Independent Subspace (IS) Loss.**    We can substitute the orthogonalisation procedure by using an auxiliary loss component for maximising the rank of the predicted subspace. For this purpose, we chose the squared cosine similarity computed between pairs of predicted vectors. The final expression for the *independent subspace* (IS) loss is

$$\text{IS Loss} = \frac{1}{K^2} \sum_{k=1}^{K} \sum_{l=1}^{K} (\mathbf{x}_k^T W \mathbf{x}_l)^2 - \frac{1}{K^2} \sum_{k=1}^{K} \sum_{l=1}^{K} (\mathbf{y}_k^T W^{\frac{1}{2}} \mathbf{x}_l)^2, \tag{9}$$

where the predictions $\{\mathbf{x}_l\}$ are normalised prior to the loss computation such that $\mathbf{x}_l^T W \mathbf{x}_l = 1$ and the scaling factor $K^2$ ensures that the loss ranges between 0 and 1. Appendix A analyses the stability of this formulation.

# 4 ARCHITECTURE

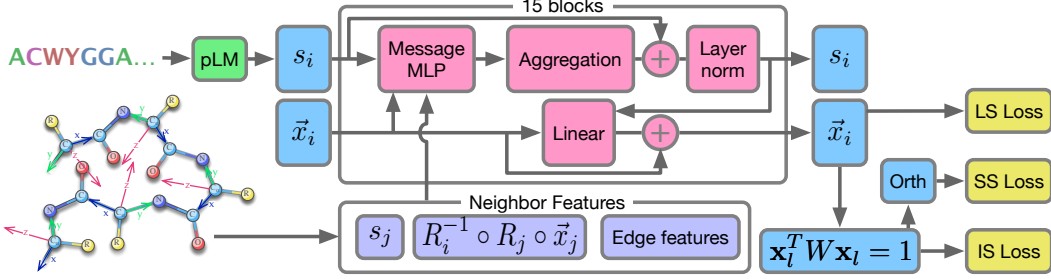

Figure 1: **PETIMOT's architecture overview.** The model processes sequence embeddings ($s$) and motion vectors ($\vec{x}$) through 15 message-passing blocks. The GNN topology and edge features are defined from the input 3D coordinates. Edge features encode 3D geometrical properties such as the relative spatial relationships between residue pairs. Each block updates both $s$ and $\vec{x}$ representations by aggregating information from neighboring residues. SE(3) equivariance is achieved by computing the features of neighbors $j$ in the reference frame of the central residue $i$. Three types of losses (LS, SS, and IS) are computed, with prior normalization of the predictions for the IS and SS losses, and an additional orthogonalisation of the predictions for the SS loss.

We solve the problem formulated above with a pLM-informed SE(3)-equivariant graph neural network called PETIMOT (Fig. 1). PETIMOT takes as input a protein sequence of length $N$, converted into an embedding $\mathbf{s}$ by a pre-trained pLM, along with 3D coordinates, and outputs a set of linear motions $X \in \mathbb{R}^{3N \times L}$.

**Dual-track representation.** PETIMOT processes protein sequences through a message-passing neural network that simultaneously handles residue embeddings and motion vectors in local coordinate frames (Fig. 1). For each residue $i$, we define and update a node embedding $\mathbf{s}_i \in \mathbb{R}^d$ initialized from pLM features and a set of $K$ motion vectors $\{\vec{x}_{ik}\}_{k=1}^{K} \in \mathbb{R}^{3 \times K}$ initialized randomly. The message passing procedure is detailed in Algorithm B.1 of Appendix B.2. The protein is represented as a graph where nodes correspond to the residues, and edges capture spatial relationships. We connect each residue $i$ to its $k$ nearest neighbors based on $C\alpha$ distances in the input structure and $l$ randomly selected residues. This hybrid connectivity scheme ensures both local geometric consistency and global information flow, while maintaining sparsity for computational efficiency. Indeed, our model scales *linearly* with the length $N$ of a protein. In our base model we set $k = 5$ and $l = 10$.

**Node and edge features.** We chose ProstT5 as our default pLM for initialising node embeddings (Heinzinger et al., 2023). This structure-aware pLM offers an excellent balance between model size – including the number of parameters and embedding dimensionality – and performance (Lombard et al., 2024b). Each residue's backbone atoms (N, CA, C) define a local reference frame through a rigid transformation $T_i \in SE(3)$. For each residue pair $(i, j)$, we compute their relative transformation $T_{ij} = T_i^{-1} \circ T_j$ from which we extract the rotation $R_{ij} \in SO(3)$ and translation $\vec{t}_{ij} \in \mathbb{R}^3$. Under global rotations and translations of the protein, these relative transformations remain invariant. Edge features $e_{ij}$ provide an SE(3)-invariant encoding of the protein structure through relative orientations, translational offsets, protein chain distance, and a complete description of peptide plane positioning captured by pairwise backbone atom distances. See Appendix B.3 for more details.

## 5 RESULTS

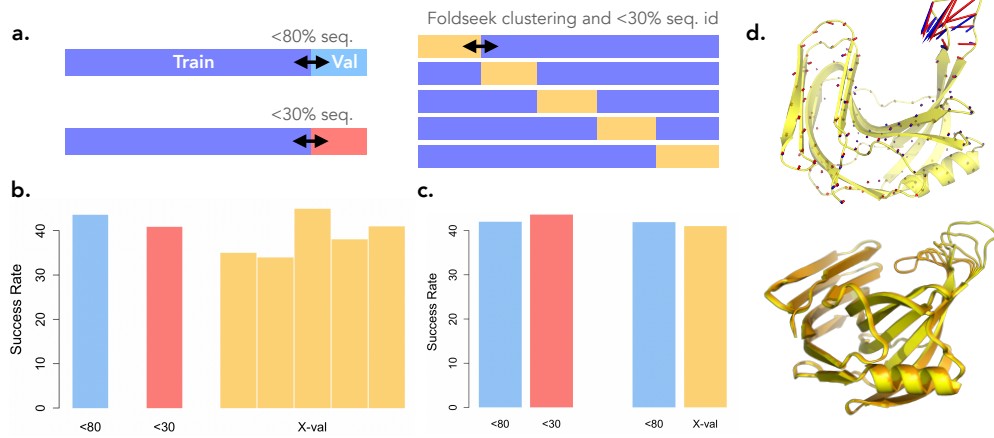

Figure 2: **Training, generalisation and prediction visualisation. a.** Training schemes. **b.** Success rates on the test sets for PETIMOT's *default*, *stringent* and *5folds* models. **c.** Left: Comparison of the *default* and *stringent* models on 734 proteins with less than 30% sequence similarity to both models' training sets. Right: Comparison between *default* and *5folds* on the 473 proteins whose structures and sequences are dissimilar to any train protein. **d.** Prediction for *B. subtilis* xylanase A (PDB id: 3EXU, chain A). On top, the predicted and ground-truth vectors are in blue and red, respectively (LS= 0.15). At the bottom, trajectory generated by deforming the protein structure along the predicted motion.

**Training and evaluation.** We trained PETIMOT against linear motions extracted from all $\sim$750,000 protein chains from the PDB (as of June 2023) clustered at 80% sequence identity and coverage (Fig. 2a). Our full training data comprises 7 335 conformational collections, which we augmented by computing the motions with respect to 5 reference conformations per collection. As a result, the full training set comprises 36,675 samples. This reference dataset encompasses conformations solved by multiple experimental techniques, including 56,866 Cryo-EM structures (30.5%) and 2,187 NMR structures (about 1.5%). This ensures representation of diverse conformational states beyond

those accessible to X-ray crystallography. We set the numbers of predicted and ground-truth motions, $K = L = 4$. See Appendices B.1 and B.4 for more details. At inference, we consider $w_i = 1$, $\forall i = 1..N$. We rely on four main evaluation metrics aimed at addressing the following questions: **1)** Is PETIMOT able to approximate at least one of the main linear motions of a given protein? For this, we rely on the minimum LS error over all possible pairs of predicted and ground-truth vectors. A prediction with LS$\leq 0.2$ almost perfectly superimposes to the ground-truth motion (Fig. 2d). We consider predictions with LS$> 0.6$ as inaccurate as they typically miss or indicate completely wrong directions for a large part of the residues involved in the motion. By comparison, the LS errors computed for random predictions are typically above 0.9; **2)** To what extent does PETIMOT capture the main motion linear subspace of a given protein? For this, we use the global SS error; **3)** Is PETIMOT able to identify the residues that move the most? Here, we rely on the magnitude error, $\frac{1}{N} \sum_{i=1}^{N} (\|\vec{y}_{ik}\|^2 - \|c_{kl}\vec{x}_{il}\|^2)$. **4)** Can PETIMOT be used to generate conformations resembling experimentally resolved functional protein states? For this, we generate conformations by deforming the input protein structure along the predicted motions and compute their RMSD to five diverse conformations selected from the ground-truth collection. See Appendix B.5 for more details.

Table 1: **PETIMOT's performance on the test set and comparison with other methods.** PETIMOT-*default* is compared with AlphaFlow, BioEmu, and the NMA on 824 test proteins. In motion subspace comparison, Min. stands for the best matching pair of predicted and ground-truth vectors. OLA refers to the optimal linear assignment between all predicted and ground-truth vectors. RMS Fluctuation (RMSF) correlation was computed between conformational samples we generated from the ground-truth or predicted motions. The Min. RMS Deviations (RMSD) were computed between each of five experimental structures and the predicted ensembles. For AlphaFlow and BioEmu, we considered their original predicted ensembles as well as the ensembles we generated by deforming the test proteins along their motion subspaces (see Appendix B.5). Arrows indicate whether higher ($\uparrow$) or lower ($\downarrow$) values are better, best results highlighted in **bold**.

| Metrics | PETIMOT | AlphaFlow | BioEmu | NMA |
|---|---|---|---|---|
| Running time $\downarrow$ | **15.82s** | 38h 07min | 39h12min | 43.59s |
| **Motion subspace comparison** | | | | |
| Success Rate (%) $\uparrow$ | **43.57** | 31.80 | 31.34 | 24.88 |
| Min. LS Error $\downarrow$ | **0.61 $\pm$ 0.22** | 0.68 $\pm$ 0.21 | 0.68 $\pm$ 0.20 | 0.72 $\pm$ 0.20 |
| Min. Magnitude Error $\downarrow$ | **0.21 $\pm$ 0.12** | 0.24 $\pm$ 0.12 | 0.23 $\pm$ 0.12 | 0.27 $\pm$ 0.14 |
| OLA LS Error $\downarrow$ | **0.83 $\pm$ 0.10** | 0.86 $\pm$ 0.10 | 0.86 $\pm$ 0.10 | 0.88 $\pm$ 0.10 |
| OLA Magnitude Error $\downarrow$ | **0.41 $\pm$ 0.14** | 0.43 $\pm$ 0.14 | 0.42 $\pm$ 0.13 | 0.48 $\pm$ 0.15 |
| Global SS Error $\downarrow$ | **0.73 $\pm$ 0.14** | 0.78 $\pm$ 0.14 | 0.77 $\pm$ 0.14 | 0.79 $\pm$ 0.14 |
| **Conformational ensemble comparison** | | | | |
| RMSF Correlation $\uparrow$ | **0.60 $\pm$ 0.23** | 0.52 $\pm$ 0.27 | 0.53 $\pm$ 0.28 | 0.48 $\pm$ 0.28 |
| Coverage, Max. Min. RMSD<2.5Å (%) | 25.70 | 24.54 | 24.39 | 22.11 |
| *original ensemble* | | *25.64* | **28.90** | |
| Avg Min. RMSD (Å) $\downarrow$ | **3.23 $\pm$ 2.51** | 3.41 $\pm$ 2.72 | 3.35 $\pm$ 2.62 | 3.50 $\pm$ 2.77 |
| *original ensemble* | | *4.77 $\pm$ 4.99* | *4.29 $\pm$ 4.34* | |

**Robustness and generalisation.** We tested PETIMOT's generalisation capabilities using three training protocols (Fig. 2a, see also Appendix B.4). In two of them, we randomly split the reference dataset into 70% for training, 15% for validation and 15% for test, where any test protein has less than 80% (*default*) or 30% (*stringent*) sequence similarity to the train proteins. In addition, we conducted 5-fold cross-validation ensuring that each fold's training set did not contain any protein chain sharing significant structural or sequence similarity with the test set (*5folds*). This protocol strictly prevents data leakage and provides robust evaluation across our complete dataset. PETIMOT's performance is robust and generalizable across the different data partitions, with success rates, defined as the fractions of test proteins with min LS$\leq 0.6$, in the 35-45% range (Fig. 2b-c, Tables 1 and C.1). See Appendix C for more details.

**Biological relevance.** To assess the biological relevance of PETIMOT's predictions, we focused on three case studies: open-closed transitions, fold switches, and multi-state cryo-EM resolved structures. For open-closed transitions, we considered the well-established iMod benchmark (Lopéz-Blanco et al., 2011) comprising a couple of tens of proteins with a wide variety of motions (hinge, shear, allosteric, and complex motions) often associated with ligand or partner binding. PETIMOT-*5folds* predicted these transitions with high accuracy, achieving a 86% success rate with an average min LS error of $0.41 \pm 0.18$, average min magnitude error $0.14 \pm 0.07$. For fold switches, we compiled a dataset of six metamorphic proteins from (Wayment-Steele et al., 2023). PETIMOT-*5folds* achieved a success rate of 37% on these challenging cases, with a min LS error of $0.67 \pm 0.17$, min magnitude error $0.25 \pm 0.14$. Our approach performed particularly well on KaiB, also highlighted in (Wayment-Steele et al., 2023). The min LS error is 0.45 starting from the ground state (2QKEC) and 0.57 starting from the FS state (5JYTA). Finally, we considered the ATPase NSF whose experimental structures correspond to ATP/ADP-bound states and 20S supercomplex conformations from cryo-EM studies (Zhao et al., 2015; White et al., 2018). The functionally relevant motions involve large-amplitude rigid-body domain movements and loop rearrangements. The first linear PCA mode explains 57% of the variance and 4 modes are required to explain 90%. PETIMOT successfully captures this complex motion subspace with min LS error as low as 0.32 and global SS error of 0.30, demonstrating its ability to predict not just single motions but biologically meaningful motion subspaces.

**Comparison with other methods.** We primarily compare PETIMOT with the NMA, a cost-effective approach for predicting the motion directions energetically accessible to a protein 3D structure. PETIMOT outperformed the NMA according to all evaluation metrics (Tables 1 and C.1). PETIMOT produced acceptable predictions for almost 40% of the dataset while the NMA's success rate is 25%, and PETIMOT achieved lower errors than the NMA in two thirds of the proteins. The NMA success cases are enriched in collective motions and depleted in localised motions. PETIMOT does not share this limitation and tends to approximate localised motions better than the NMA. Furthermore, PETIMOT was 2.75 times faster at inference (Table 1). We considered the flow-matching or diffusion based generative models Alpha/ESM-Flow and BioEmu as additional baselines. AlphaFlow was trained solely on the PDB while BioEmu was trained on massive amounts of experimental structures, 3D models, and MD conformations. To ensure fair comparison, we relied on both our motion-specific metrics and on commonly used metrics for comparing conformational ensembles (see Appendix B.5). PETIMOT outperforms both ensemble-based methods on motion subspace metrics, with a 43.57% success rate versus 31% for AlphaFlow and BioEmu, and a substantially lower Global SS Error (0.73 vs 0.77-0.78, Table 1). For the flow-matching baselines, the ESM-based version displayed slightly lower performance, compared to the AlphaFold-based one (Fig. C.2a-b). Cases where PETIMOT produces highly inaccurate predictions (min LS loss above 0.7) while the baselines are clearly successful (min LS loss below 0.4) are extremely limited (less than 5 per baseline, see for instance Fig. C.7). See Appendix C for more details.

Beyond predicting linear motions, PETIMOT allows straightforwardly generating conformational ensembles or trajectories by deforming the input protein 3D structure. We showcase this functionality on the xylanase A from *Bacillus subtilis* (Fig. 2d). We used PETIMOT predictions to generate physically realistic conformations representing the open-to-closed transition of xylanase A thumb. More broadly, the generated conformational ensembles display RMSF Pearson correlation of $0.60 \pm 0.23$ against ensembles derived from ground-truth motions, outperforming AlphaFlow ($0.52 \pm 0.27$), BioEmu ($0.53 \pm 0.28$), and NMA ($0.48 \pm 0.28$). These results demonstrate that motion subspace quality—as measured by our specialized metrics—directly translates to better prediction of conformational flexibility. Furthermore, PETIMOT conformational ensembles display the lowest average minimum RMSD ($3.23 \pm 2.51$ Å) against experimental functional states (Table 1 and Fig. C.2c-d). Notably, when all methods use identical Gaussian sampling protocols from their predicted motion subspaces, PETIMOT's superior motion quality translates directly to better conformational coverage (25.70% vs 22-24% for baselines). BioEmu's original ensemble achieves higher coverage (28.90%) by leveraging MD-informed sampling and at the expense of $\sim$3.4 seconds per conformation, demonstrating the expected trade-off between computational cost and sampling sophistication.

**Comparison of problem formulations.** Our base model combining the LS and SS loses with equal weights outperforms all three individual losses, LS, SS, and LS (Fig. B.3 and B.4). It strikes an excellent balance between approximating individual motions with high accuracy (Fig. B.3a) and globally covering the motion subspaces (Fig. B.3b). By comparison, the SS and IS losses tend

to underperform on individual motions while the LS loss tends to provide lower coverage of the ground-truth subspaces. See Appendix B.6 for more details.

**Contribution of sequence and structure features.** We performed an ablation study to assess the contribution of sequence and structure information to our architecture. Our results show that ProstT5 slightly outperforms the more recent and larger pLM, ESM-Cambrian 600M (ESM Team, 2024) (Fig. B.2). Geometrical information about protein structure provides the most significant contribution, as replacing ProstT5 embeddings with random numbers has only a small impact on network performance. Conversely, the network's performance without structural information strongly depends on the chosen pLM. While the structure-aware embeddings from ProstT5 partially compensate for missing 3D structure information, relying solely on ESM-C embeddings results in poor performance (Fig. B.2). Moreover, connecting each residue to its 15 nearest neighbours (sorted according to $C\alpha$-$C\alpha$ distances) in the protein graph results in lower performance compared to introducing randomly chosen edges or even fully relying on random connectivity (Fig. B.5). See Appendix B.6 for more details.

**Generalisation to MD data.** To further assess PETIMOT's robustness, we evaluated it on MD trajectories from the ATLAS dataset (Vander Meersche et al., 2024). We identified 400 protein chains common to both the ATLAS set and our dataset, providing an independent MD benchmark (see Appendix B.7). To ensure rigorous evaluation without data leakage, for each ATLAS protein chain we used the corresponding PETIMOT-*5folds* model trained on the fold where that specific chain was held out from training (ensuring no training exposure). PETIMOT-*5folds* achieved a 60% success rate on this MD data – with a min LS error $0.55 \pm 0.19$, min magnitude error $0.17 \pm 0.11$, and global SS error $0.60 \pm 0.16$. These performance metrics are significantly better than those obtained on experimental structures. Moreover, the association between min LS error and SS error is higher – Adjusted R-squared of 0.71 versus 0.60 on the PDB dataset (Fig. B.6). These results demonstrate that PETIMOT generalises to MD data without re-training nor fine-tuning.

**Limitations.** PETIMOT's relatively modest success rate may be partially explained by incomplete and biased functional state sampling in the PDB, where predicted motions through evolutionary transfer may correspond to functionally relevant conformational states that have not been structurally resolved, and experimental artifacts (*e.g.*, of crystallographic origin, or due to sequence engineering). Our working hypothesis is that a part of the conformational manifold represents functionally relevant motions. To address this challenge, we designed our training loss function specifically to evaluate submanifolds by calculating the minimum error between each reference motion and the set of predicted motions, allowing the model to capture conformational diversity while mitigating the impact of potential artefacts. By comparison, the Atlas MD trajectories represent an easier case but they are limited to equilibrium distributions of monomeric proteins and do not account for conformational changes induced by partner or ligand binding. In addition, our approach is limited to modeling protein motions as linear displacement vectors. While this approximation is sufficient to describe most of the observed conformational heterogeneity, it remains inadequate for modeling highly complex non-linear deformations. Furthermore, deforming proteins structures along linear motion direction may produce unrealistic conformations at large amplitudes. A possible solution yet to be investigated would be nonlinear extrapolation techniques widely used in molecular mechanics (Lopéz-Blanco et al., 2011; Hoffmann & Grudinin, 2017).

## 6 CONCLUSION

In this work, we have proposed a new perspective on the problem of capturing protein continuous conformational heterogeneity. Our approach directly infers compact and continuous representations of protein motions. Our comprehensive analysis of PETIMOT's predictive capabilities demonstrates its performance and utility for understanding how proteins deform to perform their functions. It shows that accurate motion subspace prediction—PETIMOT's core strength—provides a strong foundation for modeling protein functional dynamics, while offering interpretability and efficiency advantages over generative models for conformational sampling. PETIMOT-generated structures, while not being accurate in a thermodynamic sense, can help practitioners quickly assess possible dynamics or seed other workflows like heterogeneous cryo-EM reconstruction. Our work opens ways to future developments in protein motion manifold learning, with exciting potential applications in protein engineering and drug development.

**Ethics statement.** This paper is about machine learning models for structural biology. The research is entirely computational and does not involve human subjects, animals, or sensitive data. All the data is public. We do not anticipate any direct societal, ethical, or environmental risks arising from this work.

**Reproducibility statement.** We ensure the reproducibility of this work through the following key points:

- Our code and protocols are available in an anonymous repository. We will share the link in the discussion forum.
- The problem formulation is provided in section 3 with the proofs in Appendix A.
- A complete description of the data processing steps is provided in Appendix B.1.
- The architecture and the training protocol are detailed in Appendices B.2, B.3, B.4.
- Evaluation procedures, and baselines' parameters are explained in Appendices B.4, B.5, B.6.
- Additional experiments and ablation studies are detailed in Appendices B.6 and C.

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

APPENDICES

## A  INVARIANCE OF THE PROPOSED LOSSES

**Theorem A.1.** *SS Loss is invariant under unitary transformations of $X$ and $Y$ subspaces.*

*Proof.* Without loss of generality, let us assume that we apply a unitary transformation $U \in \mathbb{R}^{K \times K}$ to a subspace $X^\perp \in \mathbb{R}^{3N \times K}$, such that the result $X' = X^\perp U$, with $X' \in \mathbb{R}^{3N \times K}$, spans the same subspace as $X^\perp$, as it is a linear combination of the original basis vectors from $X^\perp$. Then, let us rewrite the SS loss as

$$\text{SS Loss} = 1 - \frac{1}{K} \sum_{k=1}^{K} \sum_{l=1}^{K} (\mathbf{y}_k^T W^{\frac{1}{2}} \mathbf{x}_l^\perp)^2 = 1 - \frac{1}{K} ||Y^T W^{\frac{1}{2}} X^\perp||_F^2. \tag{A.1}$$

As the Frobenius matrix norm is invariant under orthogonal, or more generally, unitary, transformations, $||Y^T W^{\frac{1}{2}} X^\perp U||_F^2 = ||Y^T W^{\frac{1}{2}} X^\perp||_F^2$, which completes the proof. $\qquad\square$

**Corollary A.1.1.** *The SS loss is invariant to the direction permutations in the Gram-Schmidt orthogonalization process.*

*Proof.* Let us consider two linear subspaces $X_1^\perp$ and $X_2^\perp$ resulting from the Gram-Schmidt orthogonalization of $X$, where we arbitrarily choose the order of the orthogonalization vectors. Both $X_1^\perp$ and $X_2^\perp$ will span the same subspace as $X$, and since both $X_1^\perp$ and $X_2^\perp$ are also orthogonal, one is a unitary transformation of the other, $X_2^\perp = X_1^\perp U$, which completes the proof. $\qquad\square$

**Theorem A.2.** *IS Loss is invariant under unitary transformations of $X$ and $Y$ subspaces.*

*Proof.* Following the previous proof, without loss of generality, let us assume that we apply an orthogonal (unitary) transformation $U \in \mathbb{R}^{K \times K}$ to a subspace $X \in \mathbb{R}^{3N \times K}$, such that the result $X' = XU$, with $X' \in \mathbb{R}^{3N \times K}$, spans the same subspace as $X$. Then, let us rewrite the IS loss as

$$\text{IS Loss} = \frac{1}{K^2} \sum_{k=1}^{K} \sum_{l=1}^{K} (\mathbf{x}_k^T W \mathbf{x}_l)^2 - \frac{1}{K^2} \sum_{k=1}^{K} \sum_{l=1}^{K} (\mathbf{y}_k^T W^{\frac{1}{2}} \mathbf{x}_l)^2 = \frac{1}{K^2} ||X^T W X||_F^2 - \frac{1}{K^2} ||Y^T W^{\frac{1}{2}} X||_F^2. \tag{A.2}$$

As the Frobenius matrix norm is invariant under orthogonal transformations, $||Y^T W^{\frac{1}{2}} XU||_F^2 = ||Y^T W^{\frac{1}{2}} X||_F^2$, and $||U^T X^T W X U||_F^2 = ||X^T W X||_F^2$, which completes the proof. $\qquad\square$

## B  METHODS DETAILS

### B.1  TRAINING DATA

**Conformational collections.**    To generate the training data, we utilized DANCE (Lombard et al., 2024a) to construct a non-redundant set of conformational collections representing the entire PDB as of June 2023. Wherever possible, we enhanced the data quality by replacing raw PDB coordinates with their updated and optimized counterparts from PDB-REDO (Joosten et al., 2014). Each conformational collection was designed to include only closely related homologs, ensuring that any two protein chains within the same collection shared at least 80% sequence identity and coverage. Collections with insufficient data points were excluded as we require at least 5 conformations. To simplify the data, we retained only C$\alpha$ atoms (option $-\mathtt{c}$) and accounted for coordinate uncertainty by applying weights (option $-\mathtt{w}$).

**Handling missing data.**    The conformations in a collection may have different lengths reflected by the introduction of gaps when aligning the amino acid sequences. We fill these gaps with the coordinates of the conformation used to center the data. In doing so, we avoid introducing biases through reconstruction of the missing coordinates. Moreover, to explicitly account for data uncertainty,

we assign confidence scores to the residues and include them in the structural alignment step and the eigen decomposition. The confidence score of a position $i$ reflects its coverage in the alignment,

$$w_i = \frac{1}{m} \sum_S \mathbb{1}_{a_i^S \neq \text{"X"}}, \tag{B.1}$$

where "X" is the symbol used for gaps and $m$ is the number of conformations. The structural alignment of the $j$th conformation onto the reference conformation amounts to determining the optimal rotation that minimises the following function (Kabsch, 1976; Kearsley, 1989),

$$E = \frac{1}{\sum_i w_i} \sum_i w_i (r_{ij}^c - r_{i0}^c)^2, \tag{B.2}$$

where $r_{ij}^c$ is the $i$th centred coordinate of the $j$th conformation and $r_{i0}^c$ is the $i$th centred coordinate of the reference conformation. The resulting aligned coordinates are then multiplied by the confidence scores prior to the PCA, as we explain below.

**Eigenspaces of positional covariance matrices.** The Cartesian coordinates of each conformational ensemble can be stored in a matrix $R$ of dimension $3N \times m$, where $N$ is the number of residues (or positions in the associated multiple sequence alignment) and $n$ is the number of conformations. Each position is represented by a C-$\alpha$ atom. We compute the coverage-weighted (to account for missing data, as explained above) covariance matrix as in Eq. 2. The covariance matrix is a $3N \times 3N$ square matrix, symmetric and real.

We decompose $C$ as $C = VDV^T$, where $V$ is a $3N \times 3N$ matrix with each column defining a sqrt-coverage-weighted eigenvector or a principal component that we interpret as a *linear motion*. $D$ is a diagonal matrix containing the eigenvalues. Specifically, the $k$th principal component was expressed as a set of 3D (sqrt-coverage-weighted) displacement vectors $\vec{x}_{ik}^{GT}, i = 1, 2, ...L$ for the $L$ C$\alpha$ atoms of the protein residues. To enable cross-protein comparisons, the vectors were normalized such that $\sum i = 1^L |\vec{x}_{ik}^{GT}|^2 = L$. The sum of the eigenvalues $\sum_{k=1}^{3m} \lambda_k$ amounts to the total positional variance of the ensemble (measured in Å$^2$) and each eigenvalue reflects the amount of variance explained by the associated eigenvector.

**Data augmentation.** The reference conformation used to align and center the 3D coordinates corresponds to the protein chain with the most representative amino acid sequence. To increase data diversity, four additional reference conformations were defined for each collection. At each iteration, the new reference conformation was selected as the one with the highest RMSD relative to the previous reference. This iterative strategy maximizes the variability of the extracted motions by emphasizing the impact of changing the reference.

## B.2 MESSAGE PASSING

The node embeddings and predicted motion vectors are updated iteratively according to the following algorithm.

---

**Algorithm B.1** PETIMOT Message Passing Block

---

1: **function** MESSAGEPASSING($\{\mathbf{s}_i\}, \{\vec{x}_i\}, \{\mathcal{N}eigh(i)\}, \{R_{ij}, e_{ij}\}$):
2:   # $\{\mathbf{s}_i\}_{i=1}^N$                                               ▷ Node embeddings
3:   # $\{\vec{x}_i\}_{i=1}^N$                             ▷ Motion vectors in local frames
4:   # $\{\mathcal{N}eigh(i)\}_{i=1}^N$                       ▷ Node neighborhoods
5:   # $\{R_{ij}, e_{ij}\}$                           ▷ Relative geometric features
6:     **for** $i = 1$ to $N$ **do**
7:         **for** $j \in \mathcal{N}eigh(i)$ **do**
8:             $\vec{x}_j^i \leftarrow R_{ij}\vec{x}_j$                     ▷ Project motion in frame $i$
9:             $m_{ij} \leftarrow \text{MessageMLP}(\mathbf{s}_i, \mathbf{s}_j, \vec{x}_i, \vec{x}_j^i, e_{ij})$
10:         **end for**
11:         $m_i \leftarrow \text{Mean}_j(m_{ij})$                   ▷ Aggregate messages
12:         $\mathbf{s}_i \leftarrow \mathbf{s}_i + \text{LayerNorm}(m_i)$        ▷ Update embedding
13:         $\vec{x}_i \leftarrow \vec{x}_i + \text{Linear}([\mathbf{s}_i, \vec{x}_i])$         ▷ Update motion
14:     **end for**
15:     **return** $\{\mathbf{s}_i\}_{i=1}^N, \{\vec{x}_i\}_{i=1}^N$
16: **end function**

---

## B.3 SE(3)-EQUIVARIANT FEATURES

We represent protein structures as attributed graphs. The node embeddings are computed with the pre-trained protein language model ProstT5 (Heinzinger et al., 2023). It is a fine-tuned version of the sequence-only model T5 that translates amino acid sequences into sequences of discrete structural states and reciprocally.

The edge embeddings are computed using SE(3)-invariant features derived from the input backbone, similarly to prior works (Ingraham et al., 2023; Dauparas et al., 2022; Ingraham et al., 2019). Specifically, the features associated with the edge $e_{ij}$ from node (atom) $i$ to node (atom) $j$ are:

- **Quaternion representation:** A 4-dimensional quaternion encoding the relative rotation $R_{ij}$ between the local reference frames of residues $i$ and $j$.

- **Relative translation:** A 3-dimensional vector representing the translation $\vec{t}_{ij}$ between the local reference frames.

- **Chain separation:** The sequence separation between residues $i$ and $j$, encoded as $\log(|i - j|+1)$.

- **Spatial separation:** The logarithm of the Euclidean distance between residues $i$ and $j$, computed as $\log(\|\vec{t}_{ij}\|+\epsilon)$, where $\epsilon = 10^{-8}$.

- **Backbone atoms distances:** Distances between all backbone atoms (N, C$\alpha$, C, O) at residues $i$ and $j$, encoded through a radial basis expansion. For each pairwise distance $d_{ab}$, we compute:

$$f_k(d_{ab}) = \exp\left(-\frac{(d_{ab} - \mu_k)^2}{2\sigma^2}\right), \tag{B.3}$$

where $\{\mu_k\}_{k=1}^{20}$ are centers spaced linearly in $[0, 20]$ Å and $\sigma = 1$ Å. This creates a $16 \times 20 = 320$ dimensional feature vector, as we have 16 pairwise distances ($4 \times 4$ atoms) each expanded in 20 basis functions.

## B.4 TRAINING PROCEDURE

For the *default* version, we randomly split the 7,335 conformational collections defined with DANCE into training, validation, and test sets with a ratio 70:15:15. The data augmentation procedure resulted in $5,119 \times 5 = 25,595$ training samples and $1,099 \times 5 = 5,495$ validation samples.

For the *stringent* version, we considered clusters of protein chains defined at 30% sequence identity and 80% sequence coverage using MMseqs2 (Steinegger & Söding, 2017). These clusters define distant protein families and we refer to them as clus-30 in the following. We kept the test proteins used PETIMOT-*default* as is and we removed all collections belonging to the same clus-30 clusters

as these proteins from the training and validation sets. Then, we re-defined a training-validation random split at the level of the clus-30 clusters with a 9:1 ratio. This operation ensures that any pair of training-validation, training-test or validation-test collections do not share more than 30% sequence identity. Finally, for each training or validation clus-30 cluster, we randomly drew 5 samples. This step ensures that each protein family is evenly represented in the training and validation sets.

For the *5fold* version, we conducted a 5-fold cross-validation experiment with strict similarity filtering over the full training set comprising 36,675 samples. We ensured a two-stage similarity filtering process:

1. **Structural similarity removal:** First, we used FoldSeek (Van Kempen et al., 2024) to cluster protein chains using an e-value threshold of 1e-2. Any two chains belonging to two different clusters do not share significant structural similarity.

2. **Sequential similarity removal:** We randomly partitioned the dataset into 5 folds and applied a cross-validation procedure. We implemented an additional sequence similarity-based filtering using MMseqs2. Specifically, for each fold, we removed from the training set (80% of the data) the protein chains sharing more than 30% sequence identity with any of the chains from the test set (20%).

The models were optimized using AdamW (Loshchilov & Hutter, 2019) with a learning rate of 5e-4 and weight decay of 0.01. We employed gradient clipping with a maximum norm of 10.0 and mixed precision training with PyTorch's Automatic Mixed Precision. The learning rate was adjusted using torch's ReduceLROnPlateau scheduler, which monitored the validation loss, reducing the learning rate by a factor of 0.2 after 10 epochs without improvement. Training was performed with a batch size of 32 for both training and validation sets. We implemented early stopping with a patience of 50 epochs, monitoring the validation loss. The model achieving the best validation performance was selected for final evaluation. We trained the model on a single NVIDIA A100-SXM4-80GB GPU. One epoch took about 9 minutes of real time.

### B.5 EVALUATION AND COMPARISON WITH OTHER METHODS

We primarily evaluated PETIMOT on a test set of 1 117 proteins, reduced to 824 to comply with other methods' requirements (see below). In addition, our 5-fold cross-validation training procedure allowed us to evaluate PETIMOT predictive capacity on the full dataset of 36,675 samples and systematically compare it with the NMA (Table C.1).

#### B.5.1 EVALUATION METRICS

We primarily relied on three metrics defined in the main text to evaluate PETIMOT and compare it with other methods: the LS error, the magnitude error, and the Global SS error. We computed the LS error and the magnitude error either on the best-matching pair of predicted and ground-truth motions (Min.) or on the full set of matched pairs determined through optimal linear assignment (OLA). In addition, we included several commonly used metrics for assessing conformational ensembles: RMSF correlation and minimum RMSD to experimental structures. To generate conformational ensembles, we deformed each test protein's 3D coordinates along its four predicted motions. We randomly sampled deformation amplitudes from a Gaussian distribution with variance proportional to each ground-truth motion's eigenvalue. For RMSF correlation, we applied the same sampling protocol to ground-truth motions to generate reference ensembles, then compared per-residue fluctuations between predicted and ground-truth ensembles. For experimental structure coverage, we leveraged our iterative strategy for data augmentation (Appendix B.1) to select five diverse conformations from the ground-truth test collections. Then, for each experimental structure, we computed its Min. RMSD to the closest conformation in the predicted motion-derived ensemble. We defined coverage as the fraction of test proteins for which all five structures have a Min. RMSD below 2.5 Å, a threshold commonly used for assessing conformational similarity in C-$\alpha$-only comparisons.

#### B.5.2 COMPARISON WITH THE NORMAL MODE ANALYSIS

We compared our approach with the physics-based unsupervised Normal Mode Analysis (NMA) method (Hayward & Go, 1995). The NMA takes as input a protein 3D structure and builds an elastic

network model where the nodes represent the atoms and the edges represent springs linking atoms located close to each other in 3D space. The four lowest normal modes are obtained by diagonalizing the mass-weighted Hessian matrix of the potential energy of this network. We used the highly efficient NOLB method, version 1.9, downloaded from `https://team.inria.fr/nano-d/software/nolb-normal-modes/` (Hoffmann & Grudinin, 2017) to extract the first $K$ normal modes from the test protein 3D conformations. Specifically, we used the following command

```
NOLB INPUT.pdb -c 10 -x -n 4 --linear -s 0 --format 1 --hetatm
```

We retained only the C$\alpha$ atoms and defined the edges in the elastic network using a distance cutoff of 10 Å. We evaluated the NMA using exactly the same metrics and protocols as those used for PETIMOT, since both methods output a set of linear motions.

### B.5.3 COMPARISON WITH GENERATIVE MODELS

We considered the flow-matching based frameworks AlphaFlow and ESMFlow as well as the diffusion-based Biomolecular Emulator (BioEmu) as additional baselines.

**Conformational ensemble generation with AlphaFlow and ESMFlow.** Out of a total of 1 117 proteins comprised in our test set, we excluded 293 proteins because they were too long (>450 amino acids) to be handled by AlphaFlow and ESMFlow in a reasonable amount of time using our computing resources. We downloaded the distilled "PDB" models from `https://github.com/bjing2016/alphaflow`. We executed AlphaFlow using the following command,

```
python predict.py --noisy_first --no_diffusion --mode alphafold
--input_csv seqs.csv --msa_dir msa_dir/
--weights alphaflow_pdb_distilled_202402.pt --samples 50
--outpdb output_pdb/
```

AlphaFlow relies on OpenFold (Ahdritz et al., 2024) to retrieve the input multiple sequence alignment (MSA). ESMFlow was launched using the same command with an additional `--mode esmfold` flag and its corresponding weights. We used AlphaFlow and ESMFlow to generate 50 conformations for each test protein and then we treated each ensemble as a conformational collection.

**Conformational ensemble generation with BioEmu.** We ran BioEmu through its python API by reading the input FASTA file then launching the *sample* function as

```
sample(sequence=record.seq, num_samples=50,
output_dir=f'./{record.id}')
```

BioEmu generated up to 50 conformations per test protein, with an average of $44 \pm 8$ conformations.

**Motion and conformation comparisons.** While PETIMOT predicts a set of linear motions from an input protein sequence and structure, Alpha/ESM-Flow and BioEmu predict a conformational ensemble from an input protein sequence. As a consequence, the input structure given to PETIMOT might not lie within the conformational ensemble predicted by Alpha/ESM-Flow or BioEmu and our evaluation framework needs adaptation to ensure fair comparison. Firstly, to directly compare motions with our three main metrics, we aligned all members of the Alpha/ESM-Flow or BioEmu collections to the test protein conformations, with identity coverage weights, and extracted the principal linear motions. Secondly, to compare conformational ensembles using RMSF and RMSD metrics, we considered the original ensembles outputted by Alpha/ESM-Flow or BioEmu, and, in addition, we generated new ensembles by deforming each test protein along the previously extracted principal linear motions. For this, we used the same protocol as that used for PETIMOT conformational ensemble generation.

We shall additionally mention that we did not filter or adapt our test set to the AlphaFlow and ESMFlow methods. As a consequence, there can be data leakage between Alpha/ESM-Flow and BioEmu train data and our test examples.

### B.5.4 RUNNING TIMES

The running times were measured on an Intel(R) Xeon(R) W-2245 CPU @ 3.90GHz equipped with GeForce RTX 3090 for PETIMOT, Alpha/ESM-Flow, and the NMA, and on an AMD Ryzen 9 7950X 16-Core CPU @ 5.88GHz equipped with NVIDIA RTX A6000 for BioEmu.

### B.6 ABLATION STUDIES

To understand the impact of different components on the performance of our model, we carried out ablation studies. We list them blow.

**Model architecture variations.**

- Network depth: We experimented with different numbers of message-passing layers (5 and 10 layers compared to our default value of 15 layers).
- Layer sharing: We tested a variant where all message-passing layers share the same parameters, as opposed to our default where each layer has unique parameters.
- Reduced internal embedding dimension: We tested a model with a smaller internal embedding dimension of 128 instead of the default 256.

Figure B.1 shows the evaluation of these modifications. A shallow 5-layers network underperforms on all evaluation metrics. The difference between other variants is not very significant.

**Structure and sequence information ablation.**

- Structure ablation: We removed all structural information from the model to assess the importance of geometric features and the performance with the PLM embeddings only. We did it by removing the edge attributes of the input of the message passing MLP.
- Sequence ablation: We ablated sequence information by replacing protein language model embeddings with random embeddings, testing them both with and without structural information.
- Embedding variants: We evaluated a different protein language model (ESMC-600M), both with and without structural tokens.

The evaluation results are shown in Fig. B.2. The results demonstrate that while both ProstT5 and ESM-Cambrian 600M perform similarly when combined with structural information, removing structural features leads to markedly different outcomes. ProstT5 embeddings partially compensate for the missing structural information, likely due to their structure-aware training, while relying solely on ESM-C embeddings results in poor performance.

**Problem formulation ablation.**    We analyzed different combinations of our loss terms (compared to our default balanced weights of LS + SS):

- Least Square loss (LS): Using only the LS loss (weight 1.0).
- Squared Sinus loss (SS): Using only the SS loss (weight 1.0).
- Independent Subspaces (IS): Using only the IS loss (weight 1.0).

Figures B.3 and B.4 compare three individual losses with the default option. The IS problem formulation underperforms on all the metrics but the global SS error. The default LS + SS formulation performs slightly better than those with individual loss components.

**Graph connectivity ablation.**    We investigated different approaches to constructing the protein graph:

- Nearest neighbor-only: Using 15 nearest neighbors (sorted according to the corresponding $C\alpha$-$C\alpha$ distances) without random edges.

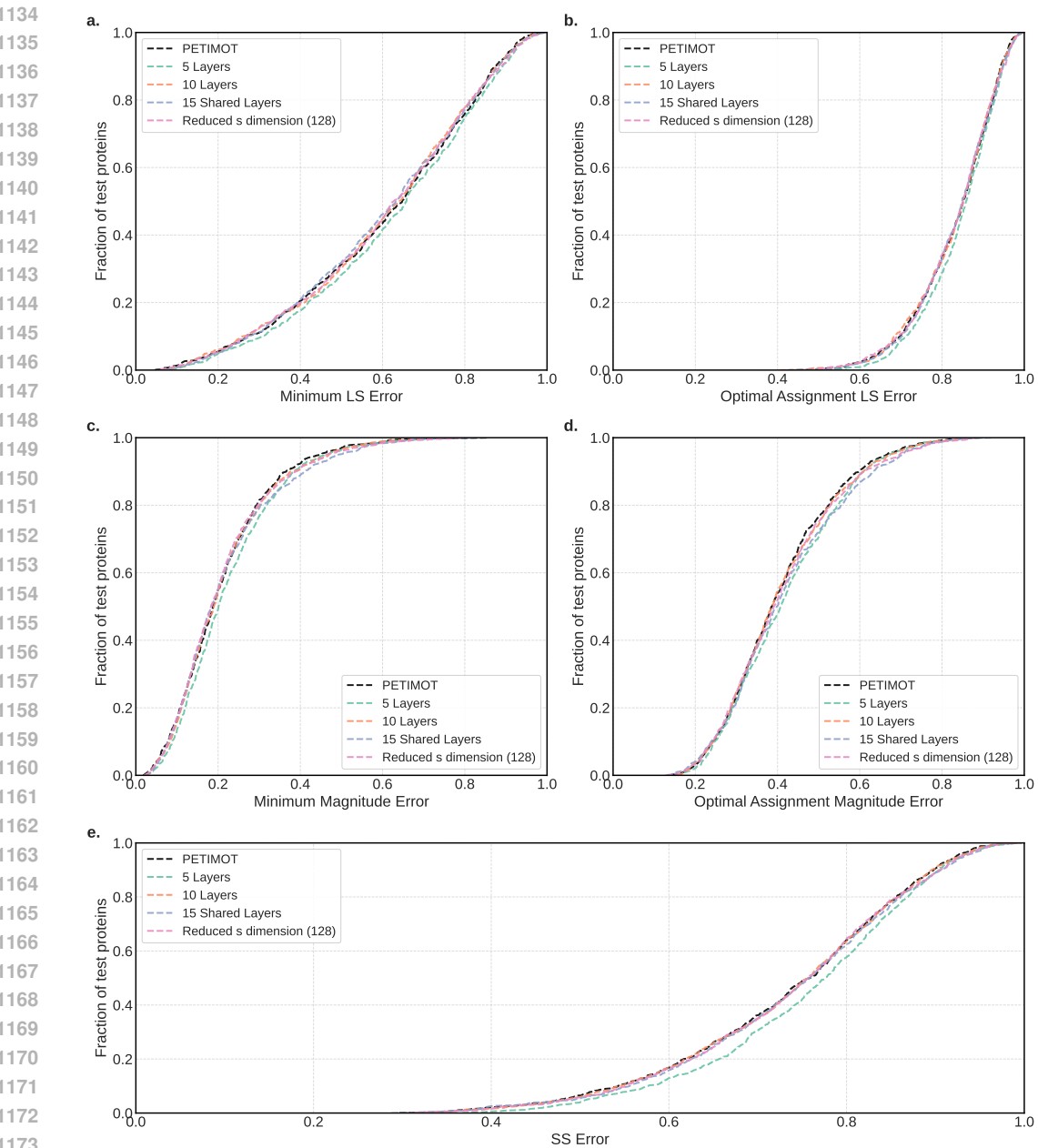

Figure B.1: **Network depth ablation.** We report cumulative curves for LS error (a-b), magnitude error (c-d), and SS error (e). For each protein, we computed the error either for the best-matching pair of predicted and ground-truth vectors (a,c) or for the best combination of four pairs of predicted and ground-truth vectors (b,d). We vary the number of layers in the network and the embedding dimension.

- Random connections-only: Using 15 random edges without nearest neighbors. This set is updated between every layer at each epoch.
- Static connectivity: Using a fixed set of random neighbors between the layers. This set is updated at each epoch.

Figure B.5 shows the ablation results. We can see that the nearest neighbor-only setup underperforms on all the metrics. Among other options, the random connectivity-only option gets lower results

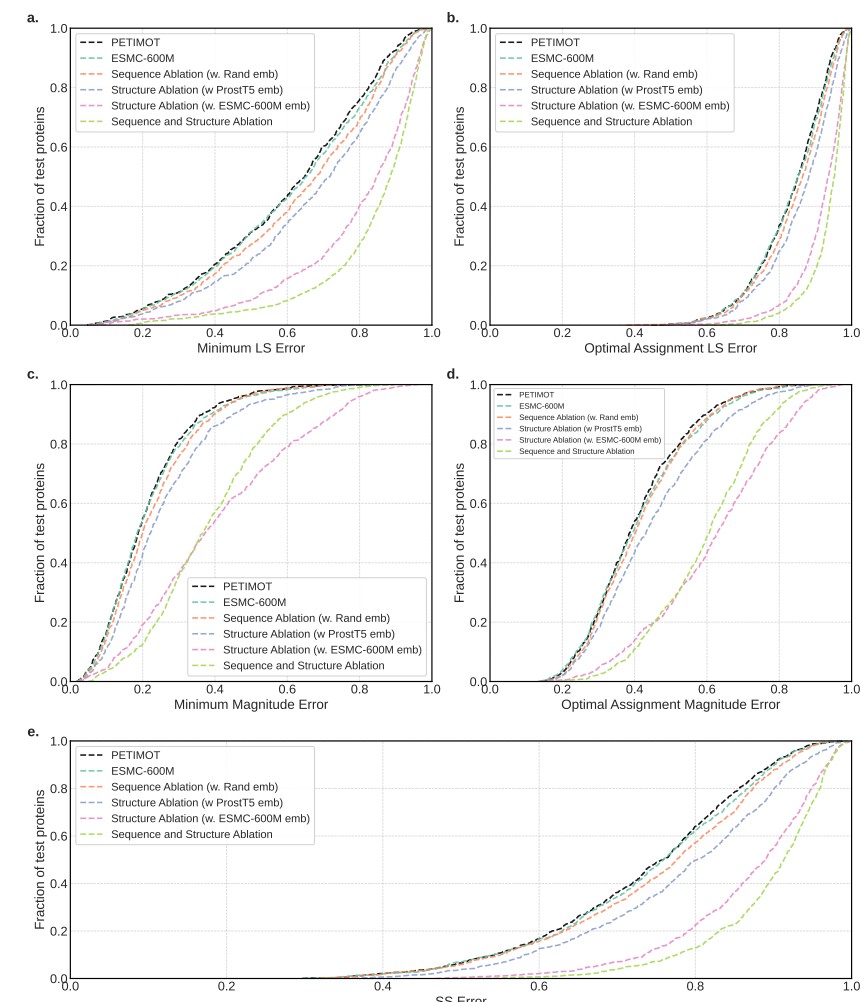

Figure B.2: **Structure and sequence information ablation study.** We report cumulative curves for LS error (a-b), magnitude error (c-d), and SS error (e). For each protein, we computed the LS and magnitude errors either for the best-matching pair of predicted and ground-truth vectors (a,c) or for the best combination of four pairs of predicted and ground-truth vectors (b,d).

at higher metrics values. The default option performs on par with the static connectivity, showing slightly better results on the optimal assignment magnitude error metrics.

B.7  ADDITIONAL VALIDATION ON ATLAS MOLECULAR DYNAMICS DATA

To further assess our model's generalization capabilities, we conducted an additional validation experiment using molecular dynamics (MD) data from the ATLAS database. This experiment provided an independent test of PETIMOT's performance on high-quality data with distinct characteristics from our training data.

The ATLAS MD dataset underwent a systematic preprocessing pipeline. First, we extracted the principal components from the MD trajectories. Subsequently, we assigned samples to appropriate cross-validation folds, ensuring strict exclusion of both structural and sequential similarity between training and test sets. This rigorous assignment process yielded 400 samples suitable for evaluation. The inference was then performed by applying our trained PETIMOT models to predict the conformational motions for each sample.

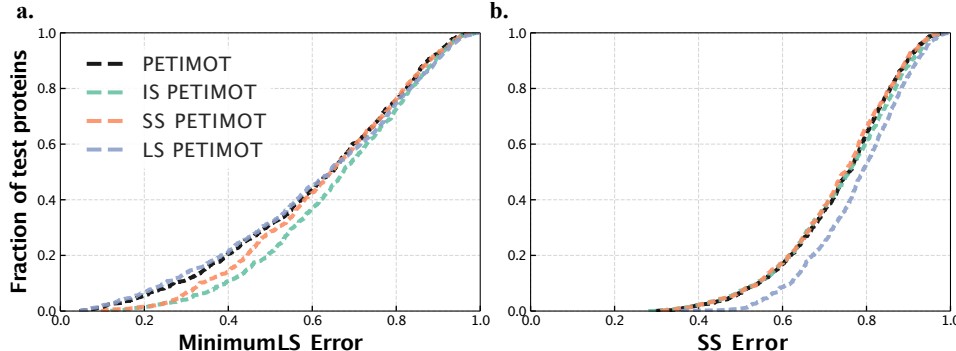

Figure B.3: **Performance comparison of different problem formulations.** We report the cumulative curve for the minimum LS error (a, best matching pair) and the global SS error (b) computed over the test set. The loss of the base model is LS + SS.

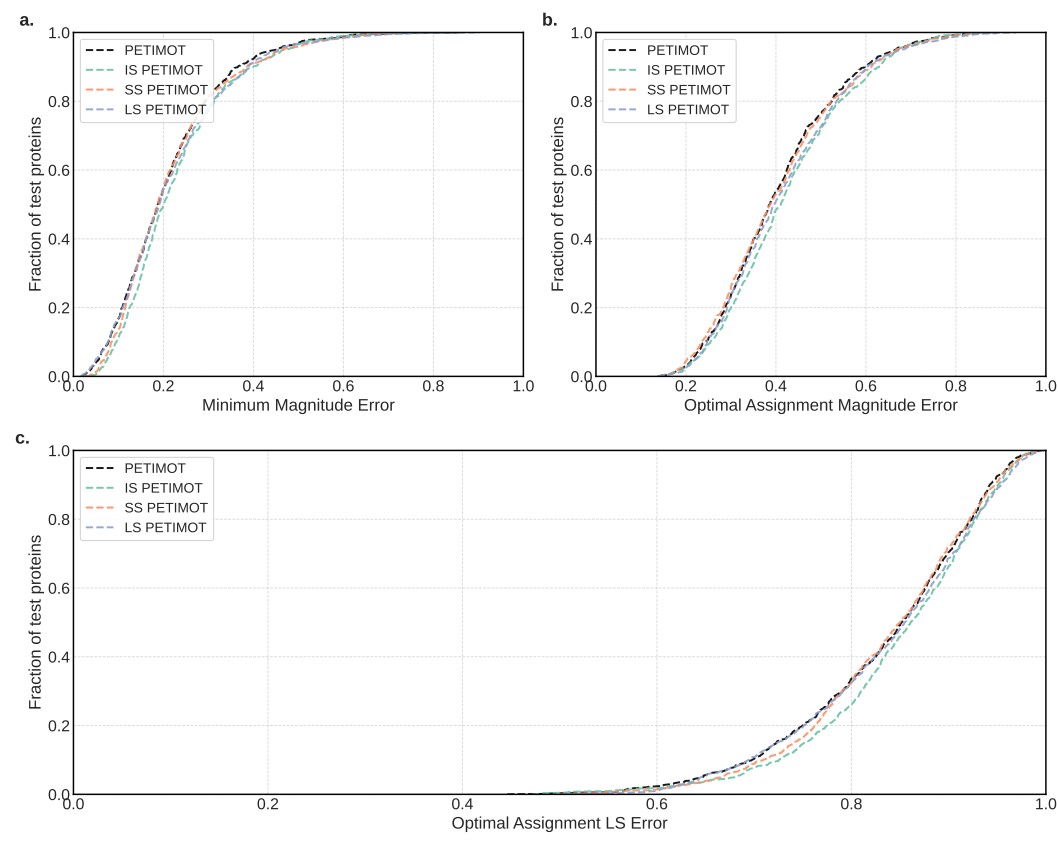

Figure B.4: **Performance comparison of different problem formulations.** We report cumulative curves for magnitude error, corresponding to the best-matching pair of predicted and ground-truth vectors (a, Min.) or the best combination of four pairs of predicted and ground-truth vectors (b, OLA) and OLA LS error (c).

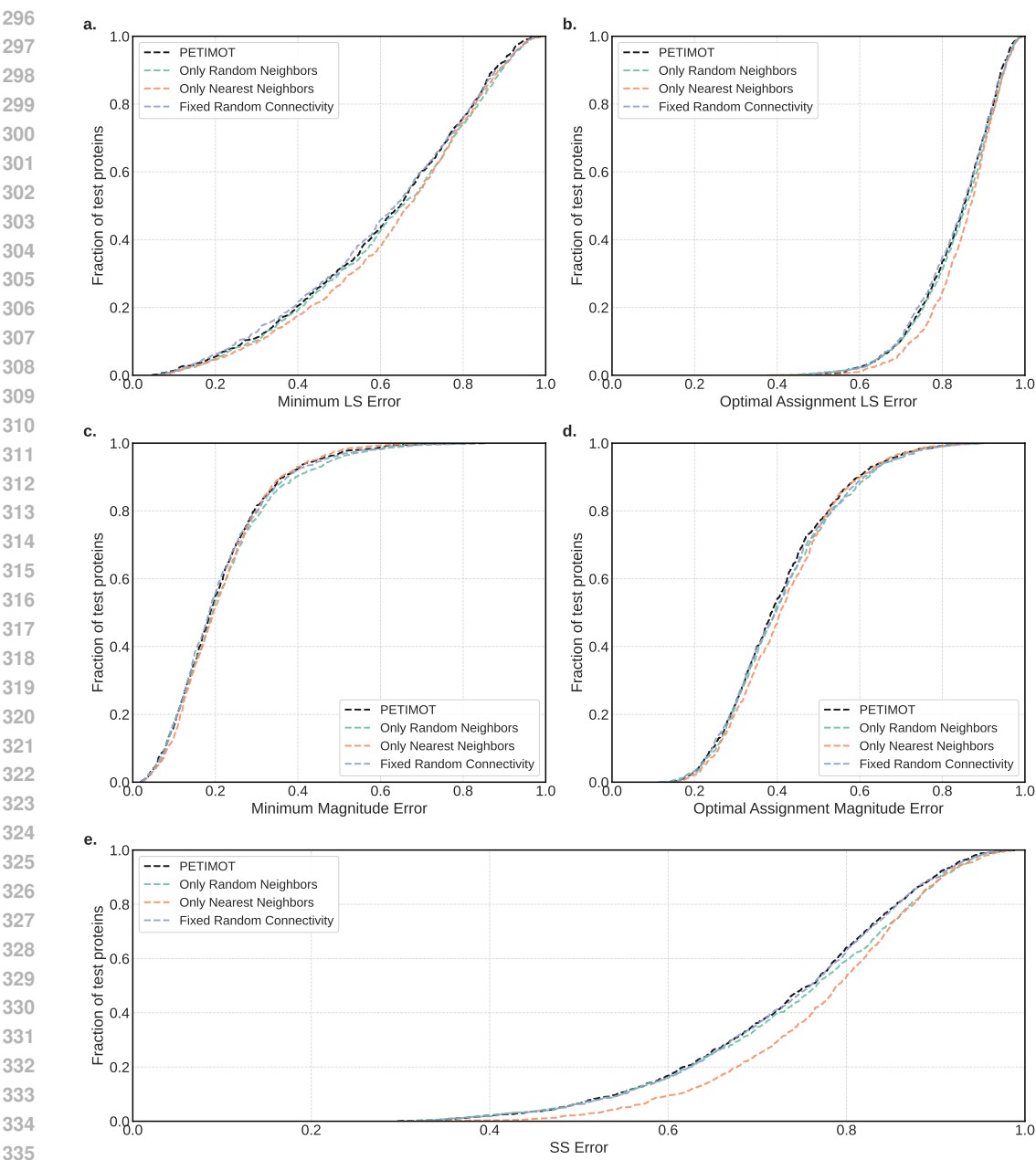

Figure B.5: **Graph connectivity ablation.** We report cumulative curves for LS error (a-b), magnitude error (c-d), and SS error (e). For each protein, we computed the error either for the best-matching pair of predicted and ground-truth vectors (a,c) or for the best combination of four pairs of predicted and ground-truth vectors (b,d). Only Random Neighbors: each residue (node) is connected to 15 randomly chosen residues and the connectivity changes after each layer. Only Nearest Neighbors: each residue (node) is connected to its 15 nearest neighbors in the input 3D structure. Fixed Random Connectivity: each residue (node) is connected to 15 residues randomly chosen at the beginning.

## C ADDITIONAL RESULTS

### C.1 GENERALISATION AND ROBUSTNESS

The performances achieved by PETIMOT-*default* on the 824 test proteins (Table 1) generalise to the full dataset with PETIMOT-*5folds* (Table C.1). The success rates, namely 43.57% and 38.98%, are

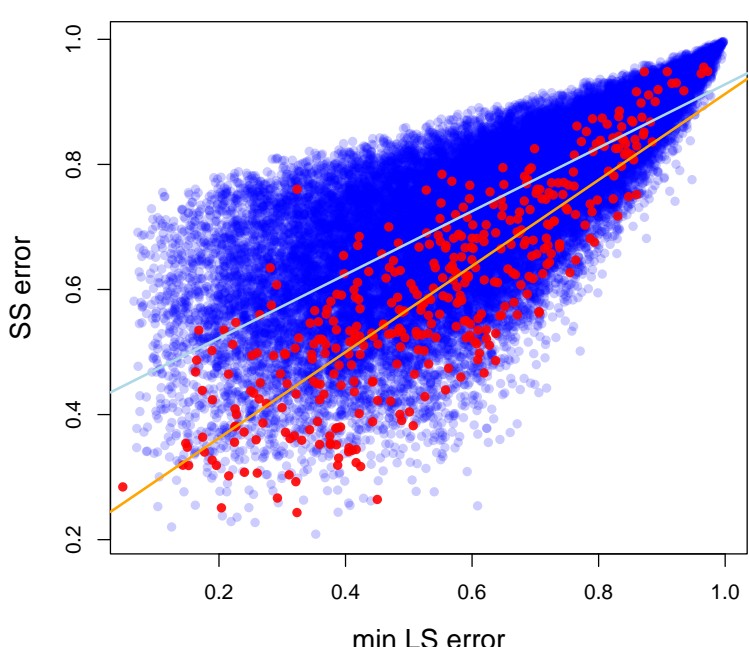

Figure B.6: **SS error in function of min LS error.** Blue dots are samples from our PDB dataset while red dots correspond to the ATLAS MD samples.

similar. Moreover, PETIMOT consistently outperforms the NMA on both the test set, and the full dataset.

Figure C.4 further assesses PETIMOT's generalisation capability across protein families. To do so, we implemented a more stringent train-validation-test split using a 30% sequence similarity threshold (see main text and Appendix B.4). When trained under this stringent protocol, PETIMOT still substantially outperforms all baseline methods according to minimum L and SS metrics (Fig. C.4a-b). Moreover, we observe a slight generalisation improvement of *PETIMOT-stringent* over *PETIMOT-default* on a test set of 474 proteins evolutionary distant from any protein used for training or validation of any of the two model versions (Fig. C.4c-d).

## C.2 COMPARISON WITH OTHER METHODS

Figure C.2 evaluates PETIMOT against Alpha/ESM-Flow, BioEmu, and the NMA using the minimum LS error, the global SS error, and the experimental structure coverage. On all the metrics we see that PETIMOT outperforms all other approaches, except for the coverage. BioEmu's original conformational ensembles cover experimental structures with lower RMSD values ($<2.5$Å) for a higher proportion of test proteins (Figure C.2d).

We also experimented with a different number of predicted components, while maintaining the number of ground-truth components fixed at $L = 4$. For these experiments, we trained additional models with the LS loss only:

- Single component prediction (1 mode).
- Reduced component prediction (2 modes).
- Extended component prediction (8 modes).

We compare these with our default setting of $K = 4$ predicted components. Figure C.2 shows the results. The key insight is that minimum-based metrics (Fig. C.2a,c) and assignment-based

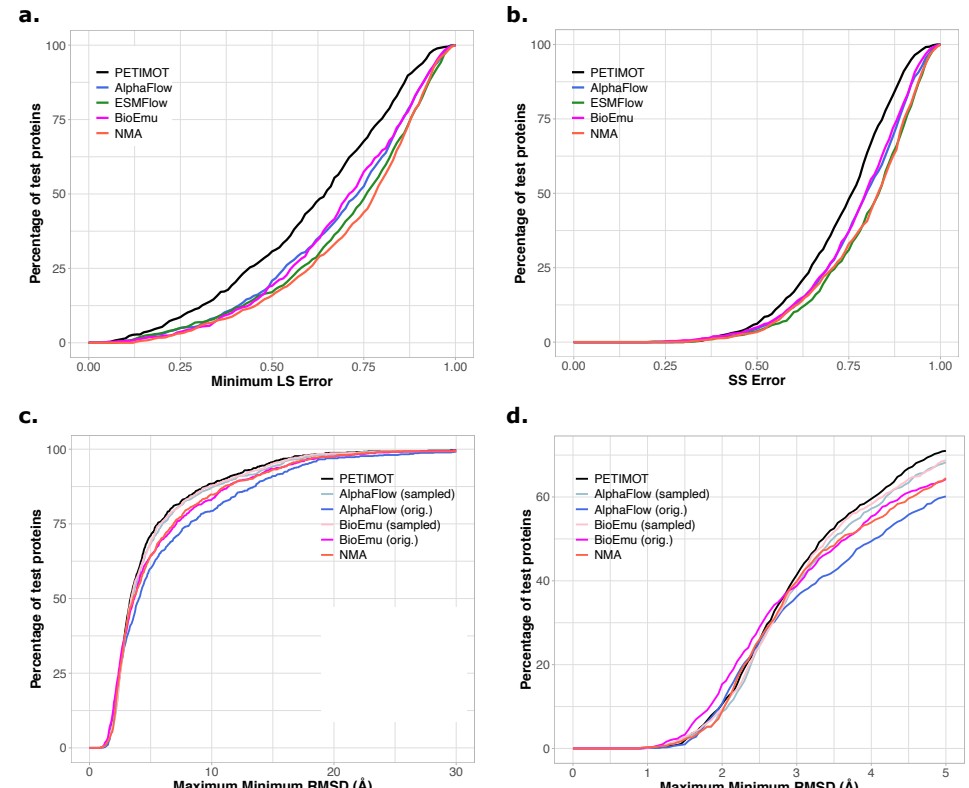

Figure C.1: **Performance comparison with other methods on the test proteins.** We report cumulative curves for minimum LS error (a), SS error (b), and experimental structure coverage (c-d). The later is computed for each test protein as the maximum value, computed over the 5 experimental reference structures, of the minimum RMSD to the predicted ensemble. For AlphaFlow and BioEmu we considered the original ensembles outputted by the method and ensembles generated by sampling along the corresponding main linear motions. Panel d shows a zoom in of the plot in panel c.

metrics (Fig. C.2b,d) measure different aspects of subspace quality: Minimum metrics measure the best possible match between any predicted and ground-truth component. These improve with more predicted components (from 1 to 8) because having more candidates increases the likelihood of finding at least one good match with each ground-truth component. Optimal assignment metrics measure overall subspace alignment by finding the best one-to-one matching between predicted and ground-truth components. Here, models with fewer predicted components (1-2) perform better

Table C.1: **Comparison of PETIMOT with the Normal Mode Analysis on the full dataset.** PETIMOT-*5folds* is compared with the NMA on 37k samples. Min. stands for the best matching pair of predicted and ground-truth vectors. OLA refers to the optimal linear assignment between all predicted and ground-truth vectors. Arrows indicate whether higher ($\uparrow$) or lower ($\downarrow$) metrics values are better. Best results are shown in **bold**.

| Metrics | PETIMOT | NMA |
|---|---|---|
| Success Rate (%) $\uparrow$ | **38.98** | 24.40 |
| Min. LS Error $\downarrow$ | **0.64 $\pm$ 0.20** | 0.72 $\pm$ 0.20 |
| Min. Magnitude Error $\downarrow$ | **0.23 $\pm$ 0.12** | 0.27 $\pm$ 0.14 |
| OLA LS Error $\downarrow$ | **0.84 $\pm$ 0.09** | 0.88 $\pm$ 0.09 |
| OLA Magnitude Error $\downarrow$ | **0.41 $\pm$ 0.13** | 0.47 $\pm$ 0.15 |
| Global SS Error $\downarrow$ | **0.75 $\pm$ 0.13** | 0.79 $\pm$ 0.14 |

because they face fewer constraints in the assignment problem - each predicted component can be matched to the best available ground-truth component without competition. The 8-component model maintains the best performance overall, as having more candidate vectors provides flexibility while still capturing the 4-dimensional ground-truth subspace effectively.

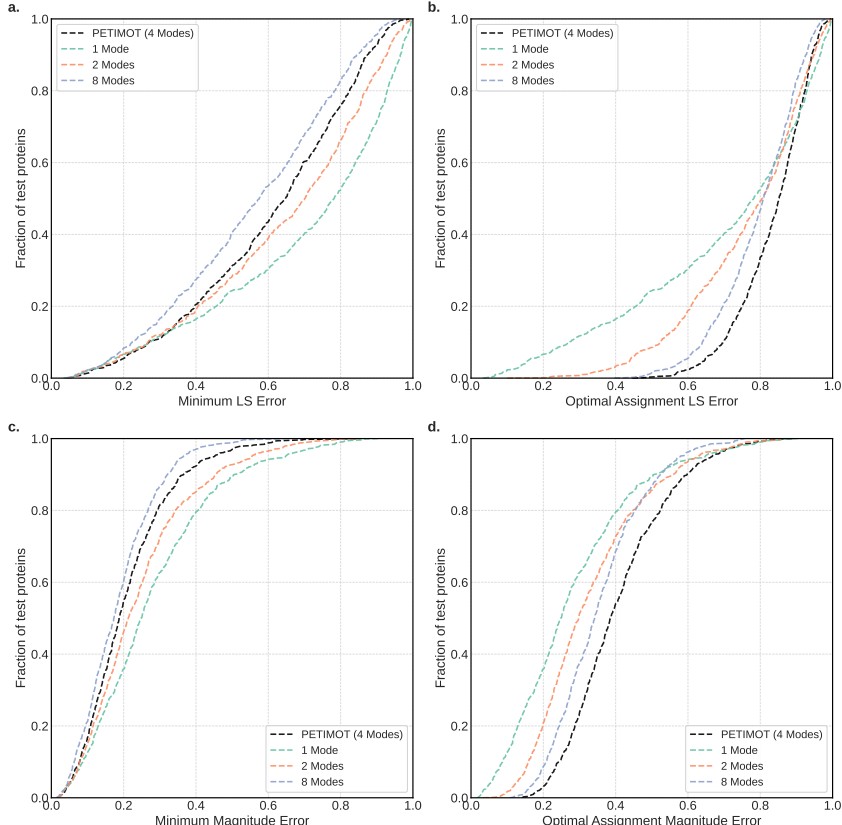

Figure C.2: **Impact of the number of predicted components.** We report cumulative curves for LS error (a-b) and magnitude error (c-d). For each protein, we computed the error either for the best-matching pair of predicted and ground-truth vectors (a,c) or for the best combination of all pairs of predicted and ground-truth vectors using optimal linear assignment (b,d). We compare models trained to predict different numbers of components (modes): 1, 2, 4, or 8, using only the LS loss.

Figure C.3 compares the accuracy of the predicted test proteins (minimum LS loss) with the structural (TM-score) and sequence (sequence identity) distances to the training set. We do not see a clear correlation between the prediction accuracy and the similarity to the training examples. Please also see Fig. 2b-c for comparison.

## C.3    VISUALISATION OF THE PREDICTIONS

Figures C.5 and C.6 show predicted (blue arrows) and ground-truth (red arrows) motion vectors for the xylanase A from *Bacillus subtilis* and the periplasmic domain of Gliding motility protein GldM from *Capnocytophaga canimorsus*, respectively.

Figure C.8 show the ground-truth main motion (yellow arrows) exhibited by experimental structures of the enzyme 2-methylisocitrate lyase (PrpB) from *Escherichia coli* and the best-matching predictions from PETIMOT (blue) and BioEmu (magenta). This enzyme, which catalyses the last step of the methylcitrate cycle, belongs to the isocitrate lyase protein family. Members of this family share an opening-closing loop mechanism associated with ligand binding. PETIMOT captures this loop motion very precisely (min LS error below 0.4) while the BioEmu ensemble mostly exhibits motions of the C-terminal helix.

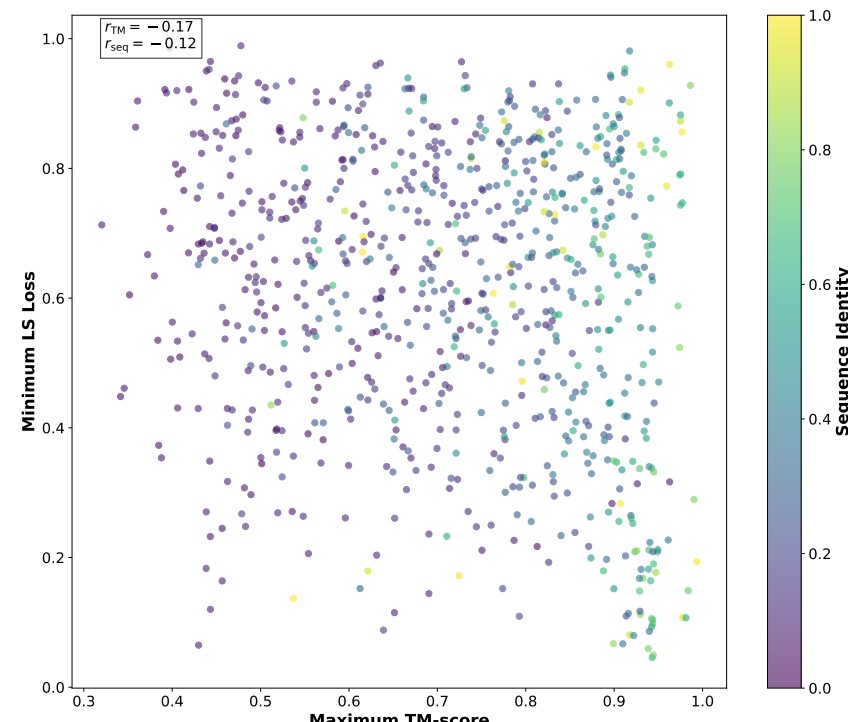

Figure C.3: **Relationship between PETIMOT's prediction accuracy and structural/sequence similarity with the training set.** The minimum LS error is plotted against the maximum TM-score between each test protein and any protein in the training set. Points are colored by the maximum sequence identity to the training samples.

# D   LICENSES FOR USED RESOURCES

In this work, we utilize several existing resources. The protein structures were obtained from the Protein Data Bank (PDB, https://www.rcsb.org/, version accessed on June 2023) which is distributed under the CC0 1.0 Universal Public Domain Dedication license (CC0 1.0). We complemented PDB data with data from PDB-redo (accessed June 2023) developed by Joosten *et al.* (Joosten et al., 2014), available at `https://pdb-redo.eu` under the licence specified at `https://pdb-redo.eu/license`. For protein language modeling, we employed ProstT5 developed by Heinzinger *et al.* (Heinzinger et al., 2023), available under the MIT license at `https://huggingface.co/Rostlab/ProstT5`, and ESM-Cambrian 600M (verison esmc-600m-2024-12) developed by the EvolutionaryScale Team (ESM Team, 2024), available under the Cambrian Non-Commercial License at `https://huggingface.co/EvolutionaryScale/esmc-600m-2024-12`. Additional resources include the DANCE method (version of Oct 8, 2024) developed by Lombard *et al.*, available under the MIT license at `https://github.com/PhyloSofS-Team/DANCE`. As baselines, we ran the NOLB method (version 1.9) developed by Hoffmann *et al.* (Hoffmann & Grudinin, 2017) and available at `https://team.inria.fr/nano-d/software/nolb-normal-modes/`, the AlphaFlow (version AlphaFlow-PDB distilled) and ESMFlow (version ESMFlow-PDB distilled) models developed by Jing *et al.* (Jing et al., 2024) and available at `https://github.com/bjing2016/alphaflow`, and BioEmu developed by Lewis *et al.* (Lewis et al., 2025) available under the MIT license at `https://github.com/microsoft/bioemu`. We used TM-align (version 20220412) developed by Zhang and Skolnick (Zhang & Skolnick, 2005) and available at `https://zhanggroup.org/TM-align/` to perform all-to-all pairwise structural alignments between train and test protein conformations and compute TM-scores.

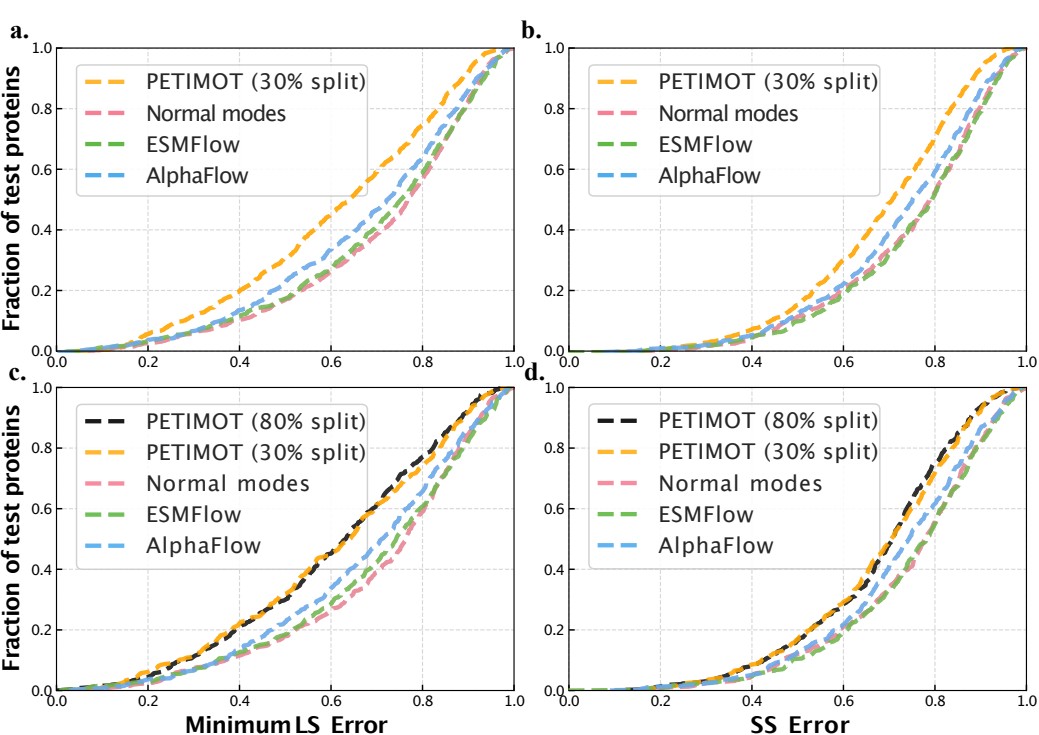

Figure C.4: **Cumulative error curves computed on the test proteins. a-b.** Comparison between *PETIMOT-stringent* model and three other methods on a non-redundant set of 734 test proteins. *PETIMOT-stringent* was trained on a stringent and non-redundant training-validation-test split defined using a 30% sequence identity threshold. **c-d.** Comparison between *PETIMOT-default*, *PETIMOT-stringent* and three other methods on 474 test proteins that share less than 30% sequence similarity with any protein used in training or validation of either model. **a,c.** Minimum LS error corresponding to the best matching pair of predicted and ground-truth motions. **b,d.** SS error computed between the entire predicted and ground-truth subspaces.

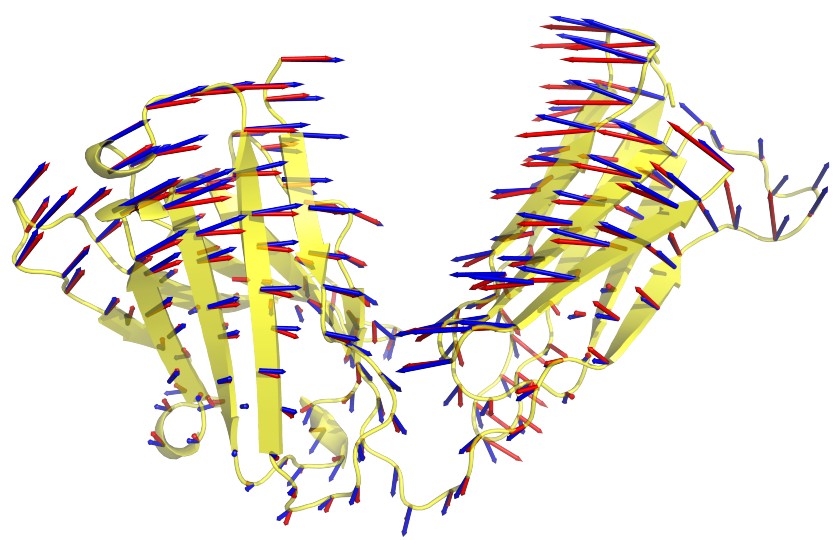

Figure C.5: **Visualization of predicted (blue arrows) and ground-truth (red arrows) motion vectors for PDB structure 3EXU (chain A), with LS error of 0.20.** The predicted deformation was used to generate the interpolated conformations shown in Fig. 2b.

Figure C.6: **Visualization of predicted (blue arrows) and ground-truth (red arrows) motion vectors for PDB structure 7SD2, with LS error of 0.18.** The predicted deformation was used to generate the interpolated conformations shown in Fig. 2c.

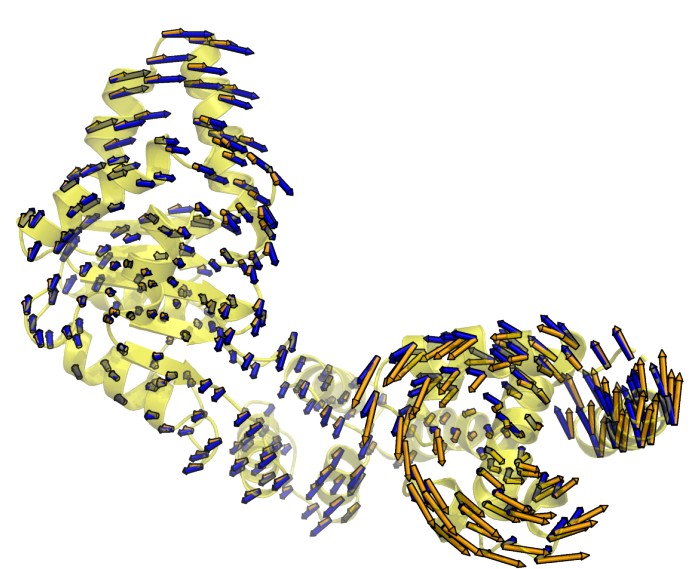

Figure C.7: **Visualization of predicted motion vectors.** PETIMOT is in blue (min LS = 0.73) and the NMA is in orange (min LS = 0.30) for the PDB structure 2HCB (chain C).

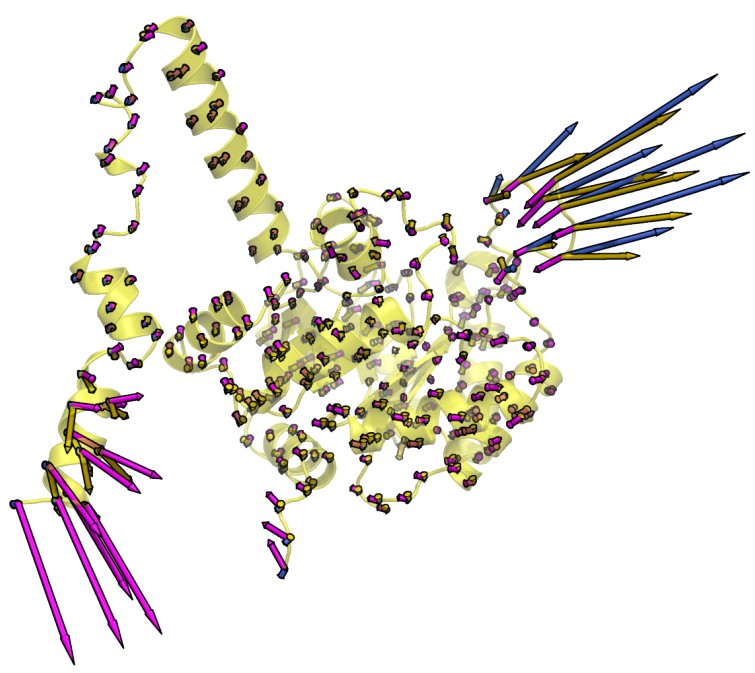

Figure C.8: **Visualization of ground-truth and predicted motion vectors for PDB structure 1OQF (chain A).** Ground-truth main motions is depicted as yellow arrows. The best-matching motions predicted by PETIMOT (fourth) and BioEmu (first) are coloured in blue and magenta, respectively.

