# OpenReview forum: "PETIMOT: A Novel Framework for Inferring Protein Motions from Sparse Data Using SE(3)-Equivariant Graph Neural Networks"
_ICLR.cc/2026/Conference — Submitted to ICLR 2026_

### Official Review · Reviewer_Dw4k · 2025-10-27

**Soundness:** 2
**Presentation:** 3
**Contribution:** 2
**Rating:** 2
**Confidence:** 4

**Summary:**

The paper proposes PETIMOT, a method to learn protein motion subspaces from sparse experimental structures using an SE(3)-equivariant GNN and embeddings from pretrained protein language models. The approach defines new subspace comparison losses (LS, SS, IS) and reports better accuracy than diffusion or physics-based models on PDB-derived datasets. However, the paper does not provide any code or data for reproducibility, which violates ICLR policy.

**Strengths:**

(1)	Addresses an interesting biological problem: modeling conformational flexibility from limited experimental data.
(2)	Combines structure-aware equivariant GNNs with protein language model embeddings in a creative way.
(3)	The proposed loss functions are mathematically clean and symmetry-aware.

**Weaknesses:**

(1)	No code or data are provided.
(2)	The evaluation is limited to PDB static structures, with no validation on dynamic or time-resolved data.
(3)	Theoretical explanations and derivations are shallow, and ablation studies are missing.
(4)	Possible redundancy or data leakage among homologous structures in the benchmark.

**Questions:**

(1)	Why wasn’t the code submitted at review time?
(2)	How do the authors ensure fair dataset splits without sequence redundancy?
(3)	Do the predicted motion subspaces correspond to known biological motions (e.g., open–closed transitions)?

---

> ### Author Response · Authors · 2025-11-16
>
> We thank the reviewer for their constructive feedback and for recognizing the biological relevance of our work and the mathematical rigor of our approach. We address each concern below:
>
> **1. Regarding code and data availability**
>
> We sincerely apologize for this oversight. Due to an internal miscommunication during the submission process, the code repository link was not included, although the repository itself was prepared and ready. The code and data are now available at: https://anonymous.4open.science/r/petimot_anonymous_iclr-CCFD/.
>
> **2. Regarding evaluation on dynamic/time-resolved data**
>
> We respectfully note that our evaluation approach is grounded in well-established methodology validated against experimental dynamic data in prior work.
>
> Our dataset construction is experimentally validated. The use of experimental structure collections to infer protein dynamics through PCA is a standard approach in structural biology (Best et al., 2006; Schneider et al., 2025; Lombard et al., 2024a; Yang et al., 2009 – cited on line 53). Crucially, Best et al., 2006 directly validates our dataset construction methodology. The authors demonstrated that dynamic properties inferred from "High Sequence-similarity Protein Data Bank (HSP) ensembles"—conceptually identical to our conformational collections—faithfully reproduce experimental NMR measurements of protein behavior in solution. They showed that even modest numbers of experimental structures sufficiently capture native conformational heterogeneity.
>
> This validation has driven methodological development. The experimental validation in Best et al., 2006 stimulated the development of computational tools (Schneider et al., 2025; Lombard et al., 2024a; Yang et al., 2009) for extracting principal modes of motion from conformational collections. Lombard et al., 2024a further demonstrated that PCA-space interpolation trajectories from experimental ATPase structures successfully recapitulate intermediate conformational states.
>
> That said, we agree that additional validation against newly acquired time-resolved data would strengthen the work. However, such datasets remain limited in scale and availability. We would welcome specific suggestions from the reviewer on accessible time-resolved datasets for systematic evaluation.
>
> **3. Regarding theoretical explanations and ablation studies**
>
> We provide theoretical derivations in Appendix A and ablation studies in Appendix B.5 (Appendix B.6 of the revision). We would appreciate more specific guidance on which theoretical aspects the reviewer finds insufficient, and we will be happy to expand on those sections. Could the reviewer clarify which theoretical explanations are missing or need deeper treatment?
>
> **4. Regarding data redundancy and leakage**
>
> We carefully addressed this concern by training models on datasets with controlled sequence and structural similarity. Specifically:
>
> - Structural similarity removal: We used FoldSeek (Van Kempen et al., 2024) with an e-value threshold of 1e-2 to ensure no significant structural similarity between clusters,
> - Sequential similarity removal: We applied MMseqs2 to remove training chains sharing >30% sequence identity with test chains,
> - We trained a 5-fold cross-validation model on these filtered subsets.
>
> Complete details are provided in Appendix B.4. This 5-fold model version demonstrates that our results hold even without homologous structures in the training set.
>
> **5. Regarding biological relevance of predicted motions**
>
> Yes, our predicted motion subspaces do correspond to known biological motions including open-closed transitions. We provide detailed analysis of biological relevance in the "Biological relevance" section (page 7), where we demonstrate correspondence with functionally important conformational changes.

---

### Official Review · Reviewer_o4hd · 2025-10-31

**Soundness:** 3
**Presentation:** 2
**Contribution:** 3
**Rating:** 6
**Confidence:** 3

**Summary:**

The paper introduces PETIMOT, a novel framework for inferring protein conformers from sparse experimental data in the PDB using SE(3)-equivariant graph neural networks. The proposed approach shifts focus from sampling full conformational distributions to learning compact linear subspaces of motions via a custom symmetry-aware loss function.

**Strengths:**

1.	The paper provides a novel formulation for the protein conformer sampling task. Given the practical importance of the task, the novel formulation is of significant value.
2.	The proposed method does not require any simulation data or physics-based guidance, being able to train utilising only sparse experimental data. The dependency on the simulated data is the severe practical limitation of existing approaches.
3.	The paper is well-structured and well-written.

**Weaknesses:**

1.	The need to introduce a novel SE(3)-equivariant architecture is unclear, given a wide range of existing equivariant graph neural networks, which could likely be adapted for this task without modifications or with minor modifications.
2.	Long-range dependencies are still approximated by random subsampling, which may fail to capture coordinated motions in large proteins; also, the process for selecting random residues lacks detail.
3.	Linear subspaces provide only limited expressibility in covering conformational ensembles, which potentially hinders the application of the method to families of proteins with highly non-linear dynamics, such as intrinsically disordered proteins or those undergoing allosteric transitions with flexible loops. At the same time, authors openly mention this limitation in the main text.
4.	Three main evaluation metrics are clear, but they seem very specific to the proposed method. Other methods like BioEmu use metrics such as relative free energy errors, flexibility correlation, and distributional similarity, while AlphaFlow includes precision, recall, and diversity via pairwise RMSD or lDDT.

**Questions:**

1.	I suggest authors firstly address weaknesses.

---

> ### Author Response · Authors · 2025-11-16
> **Answer Part 1**
>
> We sincerely thank the reviewer for their thorough evaluation and for recognizing the novelty and practical value of our formulation, as well as the quality of our presentation. We address each concern below:
>
> **1. Regarding the need for a novel SE(3)-equivariant architecture**
>
> We agree with the reviewer that our architecture builds upon existing equivariant GNN principles. Our protein representation is inspired by (Ingraham et al., 2023). However, the task we address implies specific constraints: we operate on fixed input geometries while predicting 3D motion vectors, which required careful adaptation of message-passing mechanisms to maintain equivariance throughout the prediction process. We experimented with several existing architectures and found that our message-passing formulation provided the best balance of expressiveness and computational efficiency for this specific task.
>
> **2. Regarding long-range dependencies and random subsampling:**
>
> This is a valid concern. We clarify two important points:
>
> First, information from distant residues is propagated through our repeated message-passing blocks (we use 15 blocks), which enables indirect communication across the entire protein structure even without direct long-range edges.
>
> Second, regarding the random connection strategy, we performed a graph connectivity ablation study (Appendix B.5, Fig. B4 in the original manuscript and Appendix B.6, Fig. B5 in the revision). We tested (1) Nearest neighbor-only with 15 neighbors, (2) Dynamic random connections-only with 15 random edges without nearest neighbors (sorted by Calpha-Calpha distance), updated between every layer at each epoch, and (3) Static random connections-only with 15 random edges without nearest neighbors that remain fixed from one layer to another. By comparison, the default model connects each residue with its 5 nearest neighbors and 10 randomly sampled residues excluding its nearest neighbors ; the connectivity remains fixed from one layer to another.
>
> Our ablation study result showed that the nearest neighbor-only setup underperforms on all the metrics. The default option performs on par with the static connectivity, showing slightly better results on the optimal assignment magnitude error metrics.
> We acknowledge that an all-to-all attention mechanism could potentially capture long-range interactions more directly. However, this would significantly increase computational cost, particularly for large proteins. We believe our approach provides a reasonable trade-off, though we agree that exploring attention-based long-range interactions is a valuable direction for future work.
>
> We will add more detail about the random residue selection process in the revised manuscript.
>
> **3. Regarding linear subspace limitations:**
>
> We agree this is an inherent limitation of our approach, which we openly acknowledge in the main text. The reviewer correctly identifies that highly non-linear dynamics (intrinsically disordered proteins, complex allosteric transitions with flexible loops) may not be well-captured by linear PCA subspaces.
>
> Intrinsically disordered proteins (IDPs) present a fundamentally different learning problem: they lack stable structural templates and exhibit much more diffuse conformational sampling compared to folded proteins. To quantify this difference, we performed PCA on the 28,058 MD trajectories from IDRome. On average, 20 ± 5 principal motions are necessary to explain 90% of the observed conformational heterogeneity, with the first principal motion explaining only 24.1 ± 2.4%. These values are substantially shifted from what we observe in our PDB collections.
>
> Critically, while our method learns conformational transitions from sparse, distinguishable, experimentally validated protein states, no such validation data exist for intrinsically disordered regions (IDRs). IDRome entries correspond to poorly resolved protein segments extracted from the AlphaFold database that lack proper intra-molecular constraints and cellular context (e.g., binding partners)—crucial determinants of IDR functional conformations. The associated paper (DOI: 10.1038/s41586-023-07004-5) exploits the trajectories only to derive global properties such as compaction, not to gain detailed insights into motions and conformations. From a practical perspective, the simulations are coarse-grained (only Cα atoms) and thus not adapted to our architecture. A meaningful training and evaluation of our approach on IDPs would require developing entirely different metrics and training paradigms, which goes beyond the scope of this work.

---

> ### Author Response · Authors · 2025-11-16
> **Answer Part 2**
>
> **3. Regarding linear subspace limitations (continued)**
>
>
> Linear approximations are effective for most folded proteins: While protein motions are indeed inherently nonlinear, linear approximations through PCA have proven remarkably effective for capturing dominant modes of conformational change, as extensively demonstrated in the literature. Lombard et al. (2024a) showed that in approximately half of the collections computed over the whole PDB, only one or two linear motions are sufficient to explain almost all observed conformational heterogeneity (>90% of positional variance). Moreover, the vast majority of collections (>90%, Fig. 2a) require fewer than 8 linear motions. These results demonstrate that low-dimensional linear manifolds provide a reasonable and validated means for describing most conformational diversity observed in the PDB.
>
> Identifying cases requiring nonlinear treatment: We recognize that a small subset of protein families require many linear motions (>10). This high complexity may reflect nonlinear structural deformations (e.g., those involved in allosteric transitions) or complex fluctuations. Importantly, beyond predictive performance, our approach can be used to identify such cases where the reviewer's concern about complex allosteric correlations becomes critical and warrants further investigation with nonlinear manifolds. This diagnostic capability itself represents a valuable contribution.
>
> **4. Regarding evaluation metrics:**
>
> We appreciate the reviewer's suggestion to consider alternative metrics. However, we would like to clarify that our metric choices are specifically designed to evaluate the core prediction task: learning motion subspaces, not generating conformational ensembles or predicting static structures. **Nonetheless, we have added two additional metrics in the revision. Please see updated Table 1 and sections "Comparison with other methods" and Appendix B.5. Please also see more details in the "Additional experiments" official comment above.**
>
> Our original metrics are grounded in established methodology:
> - LS error (Eq. 5 and 7): Computes the weighted pairwise least-squares difference between ground-truth and predicted motion directions. This is a direct adaptation of standard regression metrics (MAE, MSE) to the specific challenge of evaluating directional motion vectors rather than scalar values or static coordinates. We scale it between 0 and 1 for interpretability and training stability.
> - SS loss (Eq. 8): Relies on subspace coverage metrics established in the structural biology literature (Amadei et al., 1999; Leo-Macias et al., 2005; David & Jacobs, 2011). These references define the Root Mean Square Inner Product (RMSIP) as a standard measure of principal motion subspace overlap, demonstrating its effectiveness for comparing motions derived from Molecular Dynamics simulations, Anisotropic Network Models, and geometrical rigid cluster decomposition algorithms. Our SS loss is conceptually similar to RMSIP, ensuring our evaluation is grounded in validated methodology, and, more generally, to the comparison of angles between subspaces.
>
> The metrics used by BioEmu and AlphaFlow target fundamentally different prediction tasks. The reference conformational collections in our training dataset do not represent thermodynamic ensembles.  Instead, the distribution of protein states in our dataset reflects sampling biases in the PDB due to experimental conditions or to researcher’s interests. Our approach is specifically designed to cope with these biases. Indeed, the main motions extracted by PCA are those explaining the most the positional variance and thus, these calculations are not impacted by protein state relative frequencies. For instance, while collection associated with adenylate kinase comprises 35 conformations representing the closed ligand-bound state and 7 conformations for the open apo state, the main motion, explaining 99% of the variance, describe the transition between open and closed state. As a consequence, our approach is able to describe transitions between the distinct protein states that have been captured in experiments.
>
> Alternatively, one can assess the experimental structure coverage. In the revision, we evaluated the ability of predicted ensembles to cover experimentally observed conformational diversity by computing minimum RMSD to five diverse experimental structures per conformational collection (each representing a test protein and its close homologs). Please see updated Table 1 and sections "Comparison with other methods" and Appendix B.5. Please also see more details in the "Additional experiments" official comment above.

---

### Official Review · Reviewer_7oHK · 2025-11-02

**Soundness:** 3
**Presentation:** 2
**Contribution:** 2
**Rating:** 4
**Confidence:** 3

**Summary:**

In this work, the authors propose a new approach for inferring conformations of proteins as linear motions by utilizing a covariance matrix obtained after clustering pdbs by sequence similarity and extracting principle components from it. The work also proposes various loss terms to be utilized during training and evaluation.

**Strengths:**

1. Work presents an interesting approach for interpreting principle components of the defined covariance matrix as linear motions, the paper claims the conformational heterogeneity of proteins can be almost fully explained by these linear motions. Modeling protein conformations has important applications for drug discovery and enzyme engineering so this is an important research area and speed up in inferring conformational ensembles can be helpful in protein design pipelines.
2.  Table 1 shows the improved performance of PETIMOT compared to AlphaFlow on a test set.

**Weaknesses:**

There are some questions and benchmarks lacking to fully evaluate the novelty of the contribution [see below]

**Questions:**

-  R in equation 2 - Wouldn’t the size of R in this case be m x 3N not 3N x m since W is 3Nx3N
- Authors should show the linear motions mapped onto the pdbs where the PETIMOT performed poorly compared to AlphaFlow or NMA as an example (6JNA and 2HCB) and similarly for PDBs mentioned under the biological relevance to visualize the type of conformations PETIMOT does poorly on.
- The paper mentions comparing to ATLAS MD as an independent test, however its missing benchmarks beyond the values of the losses defined in the paper, authors should consider metrics like RMSD, RMSF used by AlphaFlow for comparing their generated conformation to the MD ensemble. Almost all of the results use the loss terms proposed by the paper, authors should consider metrics more commonly used by the protein modeling community like above to evaluate the true performance of the method.
- This is mentioned in the text of the paper, but for readability authors should consider including a table showing the inference time speedups.
- Bioemu was mentioned earlier in the paper when talking about related work. Why wasn't that included as a method to benchmark against as well ?

---

> ### Author Response · Authors · 2025-11-20
>
> We sincerely thank the reviewer for recognizing the practical importance of our work for drug discovery and protein engineering, and for the constructive questions that help improve our manuscript. We address each point below:
>
> **1. Regarding Equation 2 (matrix R dimensions)**
>
> We thank the reviewer for spotting this error and apologize for the confusion. In practice, we weight coordinates in advance. The correct equation should read:
> $C = W^{1/2} X X^T W^{1/2}$
> We have corrected this in the revised manuscript.
>
> **2. Regarding visualization of failure cases**
>
> We have included visualisations of PETIMOT failure cases in the revised manuscript.  Please see lines 411-413 of the revision, Figure C.7, and Appendix C. Concretely, we wrote "Cases where PETIMOT produces highly inaccurate predictions (min LS loss above 0.7) while the baselines are clearly successful (min LS loss below 0.4) are extremely limited (less than 5 per baseline, see for instance Fig. C.7). See Appendix C for more details."
>
> Additionally, we would like to stress that we conducted a systematic analysis of the similarity and differences between PETIMOT and the baselines. Overall, the differences in performance between methods are strongly correlated between the different metrics. The differences between PETIMOT and AlphaFlow are strongly correlated with those between PETIMOT and ESMFlow (Pearson R = 0.7) and also, but to a lesser extent, to those between PETIMOT and NMA (Pearson R = 0.6). This result suggests that the four methods have a similar behaviour, although they are based on very different priors and algorithms. Further supporting this observation, three quarters (76%) of PETIMOT failure cases (min LS loss above 0.6) also represent failure cases for all the baselines, AlphaFlow, ESMFlow and the NMA.
>
> **3. Regarding ATLAS MD evaluation and metrics (RMSD, RMSF)**
>
> We appreciate the reviewer's suggestion but would like to clarify why our metrics are specifically designed for our task and why standard metrics from related fields are not directly applicable.
> Our method occupies a unique position in the landscape of conformational modeling:
> Ensemble generation methods (AlphaFlow, BioEmu) produce discrete conformational samples and are evaluated with metrics like RMSD distributions, precision/recall, or thermodynamic properties. However, our method predicts continuous motion subspaces (directions + amplitudes of deformation), not discrete conformations. Generating ensembles from our subspaces would require arbitrary choices (sampling density, amplitude ranges) that would confound evaluation.
> Flexibility prediction methods (e.g., B-factor predictors, GNM/ANM-based approaches) typically predict scalar per-residue amplitudes. Our task is strictly harder: we predict both the directions and relative importance of multiple motion modes. Evaluating only amplitude correlations (like RMSF) would discard the directional information that is *central* to our contribution. **However, they can still be useful to highlight the ability of a method to identify flexible regions in a protein structure. Thus, we have included the RMSF correlation metric in the revised manuscript. Please see updated Table 1 and sections "Comparison with other methods" and Appendix B.5.**
>
> Notably, AlphaFlow itself advocates for PCA-based evaluation. The authors argue that PCA analysis provides "a more stringent test of distributional accuracy" than traditional metrics like RMSD. Our original evaluation framework aligned with this perspective: rather than comparing discrete samples, we directly evaluated the learned motion subspaces, which capture the essential geometric features of conformational distributions.
>
> **In the revised version of the manuscript, we have also added a generative setup evaluation.** Concretely, we have added the ability of predicted ensembles to cover experimentally observed conformational diversity by computing minimum RMSD to five diverse experimental structures per conformational collection (each representing a test protein and its close homologs). Please see updated Table 1 and sections "Comparison with other methods" and Appendix B.5. Please also see more details
> in the "Additional experiments" official comment above.
>
> **4. Regarding inference time comparison table**
>
> We agree this would improve readability. We have updated Table 1 showing inference time comparisons between PETIMOT, AlphaFlow, NMA, and BioEmu.
>
> **5. Regarding comparison with BioEmu**
>
> We acknowledge that BioEmu is an important state-of-the-art method that should be considered, thus we have updated the manuscript and the results accordingly.
>
> **UPDATE :** we have completed additional experiments. Please see the "Additional experiments" official comment above.

---

### Official Review · Reviewer_GHCN · 2025-11-11

**Soundness:** 1
**Presentation:** 1
**Contribution:** 1
**Rating:** 2
**Confidence:** 4

**Summary:**

This paper proposes a model of the different conformational configurations of proteins in the protein database. The model is based on a linearization of this subspace of possible conformations configurations that is based on PCA of the covariance matrix of conformations. The vectors that span this subspace of the PCA are estimated by approximately solving a linear least squares assignment problem.

**Strengths:**

Learning representations of the subspace of conformations of a protein is an important topic in proteon science.

**Weaknesses:**

The presentation of the work is not very clear. The problem is presented as a linear lead squared assignment problem (eq. 7). However it is not clear how the later described model architecture (Fig. 1) and the protein language model (line 276) relate to this task. The presentation of the paper and how the different mentioned aspects relate to each other needs to be improved.

Although I want to support protein science and diversity of applications of machine learning, it appears to be a very protein related topic with perhaps a limited audience in the ICLR community.

**Questions:**

How is the model described in section 4 used to solve the task described in section 3?

---

> ### Author Response · Authors · 2025-11-16
> **Clarification**
>
> We thank the reviewer for their feedback. We address each concern below:
>
> **1. Regarding the presentation and model architecture**
>
> We respectfully *disagree* with the assessment that the connection between Eq. 7 and our model is unclear. Let us clarify:
> - Section 3 formulates the theoretical problem: learning a linear subspace representing a set of protein functional motions,
> - Eq. 5 to 7 describe one of our loss functions, which involves determining the best-matching predicted and ground-truth motion pairs and computing a sum of squared error for each pair,
> - Section 4 and Fig. 1 describe the neural architecture that optimizes this loss.
> We establish an explicit connection between Eq. 7 and the model architecture in Fig. 1 where the “LS loss” is depicted as a yellow box.
>
> As for protein language models (pLMs): they are mentioned in the description of the architecture first on line 269 of the original manuscript and 272 in the revision (with an explicit connection with Fig. 1), where we wrote “For each residue i, we define and update a node embedding si initialized from protein language model features and a set of K motion vectors initialized randomly”. Then, on line 276 (280 in the revision), we wrote “We chose ProstT5 as our default protein language model for initialising node embeddings (Heinzinger et al., 2023)”. pLMs are widely used as feature extractors in protein prediction tasks. We use pLM embeddings as input representations, which has been shown effective for motion prediction in prior work (Lombard et al. 2024b). This motivation is explicitly stated on lines 73-74.
>
> **2. Regarding the audience and contribution**
>
> We respectfully *disagree* that protein conformational dynamics and motion prediction has "limited audience" at ICLR. Recent ICLR conferences have featured numerous protein-related papers (AlphaFold variants, protein design, dynamics prediction). Protein science is a major application domain for machine learning with significant impact.
>
> Moreover, our methodological contributions (set-based motion prediction, assignment-based losses for continuous outputs) extend beyond proteins.
>
> **3. Response to question**
>
> "How is the model described in section 4 used to solve the task described in section 3?"
> The model in Section 4 takes protein sequences/structures as input and outputs predicted motion vectors. These predicted motion vectors are referred to as “$x$” throughout the text, including in the problem formulation in Section 3 (Eq. 4) and in Figure 1. They are evaluated against ground-truth using the loss functions described in Eq. 5 to 9. For comprehensive details on the model architecture and training procedure, please refer to Appendix B.
>
> To improve clarity we added a transition between the problem formulation and the description of the architecture: "We solve the problem formulated above with a pLM-informed SE(3)-equivariant graph neural network called PETIMOT (Fig. 1). PETIMOT takes as input a protein sequence of length $N$, converted into an embedding $s$ by a pre-trained pLM, along with 3D coordinates $\vec{r}$ and outputs a set of linear motions $X \in \mathbb R^{3N \times L}$." We have also updated Figure 1 and its legend to show the input structure explicitly.

---

### Author Response · Authors · 2025-11-12
**Source code**

Dear reviewers,
We sincerely apologize for not providing access to the source code. We prepared the repository, but we simply forgot to share it. The address is: https://anonymous.4open.science/r/petimot_anonymous_iclr-CCFD/

---

### Author Response · Authors · 2025-11-25
**Additional experiments**

As suggested by several reviewers, we expanded our evaluation to include more baselines and commonly used metrics for comparing conformational ensembles.

**1) Inclusion of BioEmu as baseline.** Following reviewer 7oHK's suggestion, we included BioEmu in our benchmark. Please see updated Table 1, Figure Figure C.1, sections "Comparison with other methods" on line 396 and Appendix B.5.3 on line 1040 and C.2 on line 1388. BioEmu achieves comparable performance to AlphaFlow across all metrics (Min. LS Error: 0.68 ± 0.20, RMSF correlation: 0.53 ± 0.28). Notably, PETIMOT outperforms both ensemble-based methods on motion subspace metrics, with a 43.57% success rate versus ~31% for AlphaFlow and BioEmu, and a substantially lower Global SS Error (0.73 vs 0.77-0.78). These results reinforce our original findings and confirm that PETIMOT's direct motion prediction offers advantages over ensemble generation followed by PCA extraction.

**2) RMSF correlation analysis.** We generated conformational ensembles from predicted motion subspaces using Gaussian sampling with (ground-truth) eigenvalue-weighted coefficients, applying identical protocols to all methods for fair comparison. Please see updated Table 1 and sections "Comparison with other methods" and Appendix B.5. PETIMOT achieved an RMSF Pearson correlation of 0.60 ± 0.23, outperforming AlphaFlow (0.52 ± 0.27), BioEmu (0.53 ± 0.28), and NMA (0.48 ± 0.28). These results demonstrate that motion subspace quality—as measured by our specialized metrics—directly translates to better prediction of conformational flexibility. The performance ranking (PETIMOT > BioEmu ≈ AlphaFlow > NMA) mirrors that observed with our motion-specific metrics, validating our evaluation framework.

**3) Experimental structure coverage.** We evaluated the ability of predicted ensembles to cover experimentally observed conformational diversity by computing minimum RMSD to five diverse experimental structures per conformational collection (each representing a test protein and its close homologs). Please see updated Table 1 and sections "Comparison with other methods" and Appendix B.5. PETIMOT achieves the lowest average minimum RMSD (3.23 ± 2.51 Å) and competitive coverage at 2.5 Å threshold (25.70%). Notably, when all methods use identical Gaussian sampling protocols from their predicted motion subspaces, PETIMOT's superior motion quality translates directly to better conformational coverage (25.70% vs 22-24% for baselines). BioEmu's original ensemble achieves higher coverage (28.90%) by leveraging MD-informed sampling (~3.4 seconds per conformation), demonstrating the expected trade-off between computational cost and sampling sophistication. These results confirm that accurate motion subspace prediction—PETIMOT's core strength—provides a strong foundation for modeling protein functional dynamics, while offering interpretability and efficiency advantages over generative models for conformational sampling.

**4) Additional examples.**  We have provided additional PETIMOT prediction examples, both positive and negative, and compared them with NMA and BioEmu. Please see new Figures C.7 and C.8 in the Appendix. We wrote in the main text: "Cases where PETIMOT produces highly inaccurate predictions (min LS loss above 0.7) while the baselines are clearly successful (min LS loss below 0.4) are extremely limited (less than 5 per baseline, see for instance Fig. C.7). See Appendix C for more details." We have added the following text into Appendix : "Figure C.8 show the ground-truth main motion (yellow arrows) exhibited by experimental structures ofthe enzyme 2-methylisocitrate lyase (PrpB) from Escherichia coli and the best-matching predictionsfrom PETIMOT (blue) and BioEmu (magenta). This enzyme, which catalyses the last step of themethylcitrate cycle, belongs to the isocitrate lyase protein family.  Members of this family sharean opening-closing loop mechanism associated with ligand binding. PETIMOT captures this loopmotion very precisely (min LS error below 0.4) while the BioEmu ensemble mostly exhibits motionsof the C-terminal helix."

---

### Author Response · Authors · 2025-11-30
**Summary of the rebuttal**

We would like to take this opportunity to thank all reviewers for their constructive comments that helped us improve our manuscript and strengthen the evaluation of our method. For clarity, we provide a wrap-up of our revisions and conclusions:

- We have provided the link to the source code. We would like to apologise for this oversight during the original submission.
- We have addressed all comments from the reviewers regarding manuscript clarification.
- We have run additional experiments to extend our evaluation protocol. Specifically, we have included widely used metrics for comparing conformational ensembles, such as RMSF and RMSD. Moreover, we have added BioEmu, published in July 2025 in Science, as baseline. Our results with these new metrics and new baseline confirm that PETIMOT compares favorably with SOTA to predict protein motions and demonstrate that the conformational ensembles generated from the predicted motions capture experimentally resolved functional states.

We have updated the manuscript with changes highlighted in red.

---

### Meta-Review · Area_Chair_23sZ · 2026-01-07

**Summary:**

The authors propose PETIMOT, a SE(3)-equivariant GNN that predicts linear motion sub‑spaces for proteins directly from sequence and Cα coordinates. The model leverages ProstT5 embeddings, a 15‑layer dual‑track architecture, and a combined LS + SS loss.

**Reviewer Concerns:**

There are some positive points about the paper:
- All four reviewers acknowledge that the paper tackles an important and interesting problem in protein dynamics and that the proposed formulation (predicting linear motion sub‑spaces from sparse data) is novel.
- They all note that the method is fast (inference $\approx$ 16 s) and that it outperforms the listed baselines on the reported metrics (higher success rate, lower LS/SS errors, better RMSF correlation).
- Each reviewer raises at least one concern about the experimental evaluation (e.g., limited benchmark scope, potential data‑leakage, reliance on custom loss metrics).

- Reviewer GHCN finds parts of the presentation unclear (connection of Eq 7 to Fig 1).
- Reviewer o4hd argues against the need of a new SE(3) architecture given existing equivariant GNNs, long‑range dependencies are approximated by random subsampling with insufficient description, linear subspaces cannot capture highly non‑linear dynamics (e.g., loops, intrinsically disordered proteins), and the evaluation uses metrics specific to the proposed method while omitting other common metrics.
- Reviewer Dw4k says code and data were not released at submission time, evaluation is limited to static PDB structures with no dynamic/time‑resolved validation, theoretical derivations and ablation studies are shallow or missing, and there is a concern about possible data leakage due to homologous structures in the benchmark.

**Reviewer Scores:**

The authors's responses were:

- Clarified LS loss usage and added explanatory text linking Eq 5‑7 to Fig 1.
- Added discussion of protein dynamics as a growing ICLR topic.
- Included BioEmu in the benchmark, corrected Eq 2, and added failure‑case figures (6JNA, 2HCB).
- Expanded the methods section to justify the custom message‑passing scheme (5 nearest + 10 random edges, 15 layers) and provided ablations for edge connectivity.
- Added analysis showing that folded proteins typically need ≤ 8 PCs for 90 % variance, supporting the linear‑subspace focus.
- Released the code repository, added full derivations (Appendix A) and detailed ablations (Appendix B.5/B.6), and described redundancy removal (FoldSeek, MMseqs2) plus 5‑fold cross‑validation to address leakage concerns.
- Included standard metrics (minimum RMSD, coverage, inference‑time table) alongside LS/SS.

Overall, I feel that the paper was not quite ready for submission. More importantly, the work needs to more clearly justify why we need a new architecture for SE(3) that can't be achieved by data augmentation or some simple feature engineering.

---

### Decision · Program_Chairs · 2026-01-26

Reject